# Model Output Statistics (MOS) applied to CAMS $O_3$ forecasts: trade-offs between continuous and categorical skill scores

Hervé Petetin[1], Dene Bowdalo[1], Pierre-Antoine Bretonnière[1], Marc Guevara[1], Oriol Jorba[1], Jan Mateu Armengol[1], Margarida Samso Cabre[1], Kim Serradell[1], Albert Soret[1], and Carlos Pérez Garcia-Pando[1,2]

[1]Barcelona Supercomputing Center, Barcelona, Spain
[2]ICREA, Catalan Institution for Research and Advanced Studies, Barcelona, Spain

**Correspondence:** Hervé Petetin (herve.petetin@bsc.es)

**Abstract.** Air quality (AQ) forecasting systems are usually built upon physics-based numerical models that are affected by a number of uncertainty sources. In order to reduce forecast errors, first and foremost the bias, they are often coupled with Model Output Statistics (MOS) modules. MOS methods are statistical techniques used to correct raw forecasts at surface monitoring station locations, where AQ observations are available. In this study, we investigate to what extent AQ forecasts can be improved using a variety of MOS methods, including moving average), quantile mapping, Kalman Filter, analogs, and gradient boosting machine, and consider as well the persistence method as a reference. We apply our analysis to the Copernicus Atmospheric Monitoring Service (CAMS) regional ensemble median $O_3$ forecasts over the Iberian Peninsula during 2018-2019. A key aspect of our study is the evaluation, which is performed using a comprehensive set of continuous and categorical metrics at various time scales, along different lead times, and using different meteorological input datasets.

Our results show that $O_3$ forecasts can be substantially improved using such MOS corrections and that improvements go well beyond the correction of the systematic bias. Depending on the time scale and lead time, root mean square errors decreased from 20-40% to 10-30%, while Pearson Correlation coefficients increased from 0.7-0.8 to 0.8-0.9. Although the improvement typically affects all lead times, some MOS methods appear more adversely impacted by the lead time. The MOS methods relying on meteorological data were found to provide relatively similar performance with two different meteorological inputs.

Importantly, our results also clearly show the trade-offs between continuous and categorical skills and their dependencies on the MOS method. The most sophisticated MOS methods better reproduce $O_3$ mixing ratios overall, with the lowest errors and highest correlations. However, they are not necessarily the best in predicting the peak $O_3$ episodes, for which simpler MOS methods can achieve better results. Although the complex impact of MOS methods on the distribution and variability of raw forecasts can only be comprehended through an extended set of complementary statistical metrics, our study shows that optimally implementing MOS in AQ forecast systems crucially requires selecting the appropriate skill score to be optimized for the forecast application of interest.

# 1 Introduction

Air pollution is recognized as a major health and environmental issue (World Health Organization, 2016). Mitigating its negative impacts on health requires reducing both pollutant concentrations and population exposure. Air quality (AQ) forecasts can be used to warn the population on the potential occurrence of a pollution episode, while allowing the implementation of temporary emission reductions, including e.g. traffic restrictions, shutdown of industries and bans on the use of fertilizers in the agricultural sector).

AQ forecasting systems are typically based on regional chemistry-transport models (CTMs), which remain subject to numerous uncertainty sources, leading to persistent systematic and random errors, especially for ozone ($O_3$) and particulate matter (PM) (e.g. Im et al., 2015a, b). More importantly, they often largely underestimate the strongest episodes that exert the worst impacts upon health. In addition to the error sources related to the models themselves and the input data, part of the discrepancies between in-situ observations and geophysical forecasts are due to inherent representativeness issues, since concentrations measured at a specific location are not always comparable to the concentrations simulated over a relatively large volume.

To overcome these limitations, operational AQ forecasting systems based on geophysical models often rely on so-called Model Output Statistics (MOS) methods for correcting statistically the raw forecasts at monitoring stations. The basic idea of MOS methods is to combine raw forecasts with past observations, and eventually with other ancillary data, at a given station in order to produce a better forecast, preferably at a reasonable computational cost. As these MOS methods often significantly reduce systematic errors, bringing mean biases close to zero, they are also commonly referred to as bias-correction or bias-adjustment methods, although they may not be aimed at reducing directly this specific metric. MOS methods relying on local data (first and foremost the local observations) can also be seen as so-called downscaling methods since they allow capturing some of the local features that cannot be reproduced at typical CTM spatial resolution.

Over the last decades, several MOS methods have been proposed for correcting weather forecasts, before their more recent application to AQ forecasts, essentially on $O_3$ and fine particulate matter ($PM_{2.5}$, with aerodynamic diameter lower than 2.5 µm). A very simple approach consists in subtracting the mean bias (or multiplying by a mean ratio to avoid negative values in the corrected forecasts) calculated from past data (McKeen et al., 2005). A more adaptive version consists in correcting the forecast by the model bias calculated over the previous days, which assumes some persistence in the errors (Djalalova et al., 2010). Other authors proposed fitting linear regression models between chemical concentration errors and meteorological parameters (e.g., Honoré et al., 2008; Struzewska et al., 2016). Liu et al. (2018) applied a set of autoregressive integrated moving average (ARIMA) models to improve Community Multiscale Air Quality (CMAQ) model forecasts. The Kalman Filter (KF) method is a more sophisticated approach, yet still relatively simple to implement, based on signal processing theory (e.g., Delle Monache et al., 2006; Kang et al., 2008, 2010; Borrego et al., 2011; Djalalova et al., 2010, 2015; Ma et al., 2018). Initially employed for correcting meteorological forecasts (Delle Monache et al., 2011; Hamill and Whitaker, 2006), the ANalogs (AN) method provides an observation-based forecast using historical forecasts and has recently provided encouraging results for correcting $PM_{2.5}$ CMAQ forecasts over the United States (Djalalova et al., 2015; Huang et al., 2017).

A common limitation in the aforementioned studies is that MOS corrections are assessed mainly in terms of continuous vari-

ables (i.e. pollutant mixing ratios), while typically less attention is put on the parallel impact in terms of categorical variables (i.e. exceedances of given thresholds), which is yet one of the primary goals of AQ forecasting systems. This can give a partial, if not misleading, view of the advantages and disadvantages of the different MOS approaches proposed in the literature.

The present study aims at providing a comprehensive assessment of the impact of different MOS approaches upon AQ forecasts. We consider a representative set of MOS methods, including some already proposed in the recent literature and another one based on machine learning (ML). These MOS corrective methods are applied to the Copernicus Atmospheric Monitoring Service (CAMS) regional ensemble $O_3$ forecasts, focusing on the Iberian Peninsula (Spain and Portugal) during the period 2018-2019. The MOS methods are evaluated for a comprehensive set of continuous and categorical metrics, at various time
scales (hourly to daily), along different lead times (1 to 4 days), with different meteorological input data (forecast vs reanalyzed), in order to provide a more complete vision of their behavior.

The paper is organized as follows: Sect. 2 first describes the data and MOS methods used in this study; Sect. 3 includes the evaluation of the raw (uncorrected) CAMS regional ensemble $O_3$ forecast over the Iberian Peninsula, along with a detailed assessment of the MOS results and some sensitivity analyses; a broader discussion and conclusion are provided in Sect. 4.

## 2   Data and methods

### 2.1   Data

#### 2.1.1   Ozone observations

Hourly $O_3$ measurements over 2018-2019 are taken from the European Environmental Agency (EEA) AQ e-Reporting (EEA, 2020), and accessed through GHOST v3.2.2 (Globally Harmonised Observational Surface Treatment). GHOST is a project
developed at the Earth Sciences Department of the Barcelona Supercomputing Center that aims at harmonizing global surface atmospheric observations and metadata, for the purpose of facilitating quality-assured comparisons between observations and models within the atmospheric chemistry community (Bowdalo, in preparation). On top of the public datasets it ingests, GHOST provides numerous data flags that are here used for quality assurance screening (see Appendix A). In this study, daily mean, daily 1-hour maximum and daily 8-hour maximum (hereafter respectively referred to as d, d1max and d8max) are com-
puted only when at least 75% of the hourly data are available (i.e. 18 over 24 hours). Note that despite such data availability criteria, large data gaps at some stations and during some days might occur mainly during daytime (for instance due to maintenance operations that typically occur during working hours). Considering all stations and days with at least 18 hours of data, the frequency of data gaps exceeding 4 hours between 8 and 15 UTC was found to be only 0.6% (1854/314,005). Such situation occurs with a similarly low frequency on days exceeding the target threshold (77/13,221 or 0.6%) and never occurs on days
exceeding the information threshold.

Our study focuses on the Iberian Peninsula, over a domain ranging from 10°W to 5°E longitude and from 35°N to 44°N lati-

tude that includes Spain, Portugal and part of south-western France. In total, 455 $O_3$ monitoring stations are included, which represents an observational dataset of 7,437,862 hourly $O_3$ measurements with 93% of hourly data availability.

### 2.1.2 CAMS regional ensemble forecast

The benefit of MOS corrections is investigated on the CAMS regional ensemble forecasts. As one of the six Copernicus services, CAMS provides AQ forecast and reanalysis data at both regional and global scales (https://www.regional.atmosphere.copernicus.eu/). At regional scale, 9 state-of-the-art CTMs developed by European research institutions are currently participating in the operational ensemble AQ forecasts (CHIMERE from INERIS, EMEP from MET Norway, EURAD-IM from University of Cologne, LOTOS-EUROS from KNMI and TNO, MATCH from SMHI, MOCAGE from METEO-FRANCE, SILAM from FMI, DEHM

from Aarhus University, GEM-AQ from IEP-NRI). In addition, MONARCH from BSC and MINNI from ENEA will join the ensemble soon. The ensemble forecast is computed as the median of all individual forecasts. Note that due to possible technical failures, all 9 forecasts are not always available for computing the full ensemble. The CAMS regional forecasts are provided over 4 lead days, hereafter referred to as D+1, D+2, D+3 and D+4 (starting at 0 UTC).

### 2.1.3 HRES and ERA5 meteorological data

Some MOS methods rely on meteorological data. In this study, meteorological data are taken from the Atmospheric Model high resolution 10-days forecast (HRES) (https://www.ecmwf.int/en/forecasts/datasets/set-i) provided by the European Centre for Medium-Range Weather Forecasts (ECMWF). HRES has a native spatial resolution of about 9 km and 137 vertical levels. In addition, to investigate the sensitivity to the meteorological input data, we replicated all our experiments with the ERA5 reanalysis dataset (Copernicus Climate Change Service (C3S), 2017) (https://www.ecmwf.int/en/forecasts/datasets/

reanalysis-datasets/era5). ERA5 data have a native spatial resolution of about 31 km and 137 vertical levels, although data were downloaded on a 0.25°x0.25° regular longitude-latitude grid from the Climate Data Store. At all surface $O_3$ monitoring stations, for both HRES and ERA5, we extracted the following variables at the hourly scale: 2-m temperature (code 167), 10-m surface wind speed (207), normalized 10-m zonal and meridian wind speed components (165 and 166), surface pressure (134), total cloud cover (164), surface net solar radiation (176), surface solar radiation downwards (169), downward UV radiation at

the surface (57), boundary layer height (159), and geopotential at 500 hPa (129).

### 2.2 Applying MOS under restrictive operational conditions

A novel aspect of this study is that we provide a comparison of a set of MOS methods under potentially restrictive training conditions in operational context. To mimic such restrictions we assume that (1) no past data, neither modeled nor observed, are available for training at the beginning of the period of study (here 2018/01/01), (2) the amount of modeled and observed data

continuously grows with time along the period of study (here 2018-2019). On a given day, the MOS methods can therefore only rely on the historical data accumulated since the beginning of the period. Our approach consists in understanding the behavior of the different MOS methods in a worse case scenario where a new or upgraded operational AQ forecasting system

is implemented together with a MOS module for which there is little or no hindcast data. We believe that such a strategy allows to compare the different MOS methods in a balanced way given the operational context. As described in detail in the next section, some MOS methods require very limited prior information to achieve their optimal performance, while others need a larger amount of training data. In an operational context, the first category of methods might thus be advantaged at the beginning before being gradually supplanted with the second category. We note, however, that methods relying on limited past data may respond better to an abrupt change in environmental conditions, as experienced for instance during the COVID-19 lockdowns. Although not covered by the present study, we acknowledge here that in an operational context, the relationship between the length of past training data and the performance of the corresponding MOS prediction is an interesting aspect to investigate, as is the quantification of the spin-up time beyond which the MOS method might not significantly improve. Only some insights will be given by comparing the performance obtained in 2019 with and without using the data available in 2018. Similarly, our study does not investigates how potential issues (delays) in the near-real time availability of the observations can impact the performance of the MOS methods, although this might be another important aspect to take into account in operational conditions; to the best of our knowledge, EEA observations are typically available with a 2-h lag but some sporadic technical failures can induce extended delays.

## 2.3 Description of the Model Output Statistics (MOS) methods

This section describes the different MOS methods implemented for correcting the raw forecasts (hereafter referred to as RAW), namely: moving average (MA), Kalman filter (KF), quantile mapping (QM), analogs (AN) and gradient boosting machine (GBM). All MOS methods are applied independently on each monitoring station. The skill of these different forecasts (including the RAW) is assessed relative to the Persistence (PERS) reference method, which uses the previously observed concentration values at a specific hour of the day (averaged over 1 or several days) as the predicted value. As a first approach, we use a time window of one single day (hereafter referred to as PERS(1)).

### 2.3.1 Moving average (MA) method

We primarily consider the Moving Average (MA) method, by which the raw CAMS forecast bias in the previous day(s) is used to correct the forecast. As a first approach, we use a time window of one single day (hereafter referred to as MA(1)). The sensitivity to the time window is discussed in Sect. 3.4.

### 2.3.2 Quantile mapping (QM) method

The quantile mapping (QM) method aims at adjusting the distribution of the forecast concentrations to the distribution of observed concentrations. For a given day, the QM method consists in (1) computing two cumulative distribution functions (CDF), corresponding to past modeled and observed $O_3$ mixing ratios, respectively, (2) locating the current $O_3$ forecast in the model CDF and (3) identifying the corresponding $O_3$ values in the observation CDF and using it as the QM-corrected $O_3$ forecast. For instance, if the current $O_3$ forecast gives a value corresponding to the 95[th] percentile, the QM-corrected $O_3$

forecast will correspond to the 95th percentile of the observed $O_3$ mixing ratios. This approach thus aims at correcting all quantiles of the distribution, and not only the mean.

In the operational-like context in which this study is conducted (Sect. 2.2), first QM corrections are computed when 30 days of data have been primarily accumulated, to ensure a minimum representativeness of the model and observation CDFs. For computational reasons, both CDFs are updated every 30 days (although an update frequency of one single day would be optimal in a real operational context). The choice of a 30-day update frequency only aims at reducing the computational cost of running all MOS methods at all stations during the 2-year period. In a real operational context, only one day would have to be run, which would allow increasing the update frequency up to 1 day, i.e., the CDFs would be updated every day ensuring that we are taking benefit from the entire observational dataset available at a given time.

### 2.3.3  Kalman filter (KF) method

The Kalman Filter (KF) is an optimal recursive data processing algorithm with numerous science and engineering applications (see Pei et al. (2017) for an introduction). In atmospheric sciences, it offers a popular frame for sophisticated data assimilation applications (e.g., Gaubert et al., 2014; Di Tomaso et al., 2017), but can also be used as a simple yet powerful MOS method for correcting forecasts (e.g., Delle Monache et al., 2006; Kang et al., 2008; De Ridder et al., 2012). The KF-based MOS method aims at estimating recursively the unknown forecast bias (here taken as the state variable of interest) combining previous forecast bias estimates with forecast bias observations. The updated forecast bias estimate is computed as a weighted average of these two terms, both being considered as uncertain, i.e. affected by a noise with zero-mean and a given variance. A detailed description of the KF algorithm can be found in Appendix B but an important aspect to be mentioned here is that each of these two terms is weighted according to the value of the so-called Kalman gain that intrinsically depends on the ratio of both variances (hereafter referred to as the variance ratio). The value chosen for this internal parameter substantially affects the behavior of the KF, and thus the obtained MOS corrections. A variance ratio close to zero induces a Kalman gain close to 0. In such situations, the estimated forecast bias corresponds to the estimated forecast bias of the previous day, independently from the forecast error. A very high (infinite) variance ratio gives a Kalman gain close to 1. In this case, the estimated forecast bias corresponds to the observed forecast bias of the previous day, which makes it thus equivalent to the MA(1) method.

In this study, the variance ratio is adjusted dynamically and updated regularly in order to optimize a specific statistical metric, in our case the RMSE (the corresponding approach being hereafter referred to as KF(RMSE)). The different steps are: (1) at a given day of update, the KF corrections over the entire historical dataset are computed considering different values of variance ratio, from 0.001 to 100 in a logarithmic progression; (2) the RMSE is computed for each of the corrected historical time series obtained; (3) the variance ratio associated to the best RMSE is retained and used until the next update. Other choices of metric to optimize are explored in Sect. 3.4.

As for QM, for computational reasons, the update frequency is set to 30 days in this study (although, again, an update frequency of one single day would be optimal).

### 2.3.4 Analogs (AN) method

The analogs method (AN) implemented here consists in (1) comparing the current forecast to all past forecasts available, (2) identifying the past days with the most similar forecast (hereafter referred to as analog days or analogs), and (3) using the corresponding past observed concentrations to estimate the AN-corrected $O_3$ forecast (e.g., Delle Monache et al., 2011, 2013; Djalalova et al., 2015; Huang et al., 2017). The current forecast is compared to each individual past forecast in order to identify which ones are the most similar. Based on a set of features including the raw $O_3$ mixing ratio forecast from the AQ model and the 10-meter wind speed, 2-meter temperature, surface pressure and boundary layer height forecast from the meteorological model, the distance metric proposed by Delle Monache et al. (2011) and previously used in Djalalova et al. (2015) (see the formula in Appendix C) is used to compute the distance (i.e., to quantify the similarity) of each individual past forecast with respect to the current forecast. Then, as a first approach, the 10 best analog days that correspond here to the 10 most similar past forecasts are identified (hereafter referred to as AN(10); other values are tested in Sect. 3.4)." From those best analog days, the MOS-corrected forecast is computed as the weighted average of the corresponding observed concentrations, where weights are taken as the inverse of the distance metric previously computed. In comparison to a normal average, introducing the weights is expected to slightly reduce the dependence upon the number of analog days chosen.

Therefore, in the analogs paradigm, the past days of similar chemical and/or meteorological conditions are identified in the forecast (i.e. model) space while the output (i.e. the AN-corrected forecast) is taken from the observation space. The AQ model thus only serves to identify the past observed situations that look similar to the current one.

### 2.3.5 Machine-learning-based MOS method

We also explore the use of ML algorithms as an innovative MOS approach for correcting AQ forecasts. In ML terms, it corresponds to a supervised regression problem where a ML model is trained to predict the observed concentrations, hereafter referred to as the target or output, based on multiple ancillary variables, hereafter referred to as the features or inputs, coming from meteorological and chemistry-transport geophysical models and/or past observations. In this context, the use of ML is of potential interest because (i) we suspect that some relationships exist between the target variable and at least some of these features, (ii) these relationships are likely too complex to be modeled in an analytical way, and (iii) data are available for extracting (learning) information about them. Over the last years, ML algorithms became very popular for many types of predictions, notably due to their ability to model complex (typically non-linear and multi-variable) relationships with good prediction skills. Among the myriad of ML algorithms developed so far, we focus on the decision tree-based ensemble methods, and more specifically on the gradient boosting machine (GBM), that often gives among the best prediction skills (as shown in various ML competitions and model intercomparisons, e.g., Caruana and Niculescu-Mizil, 2005).

At each monitoring station, one single ML model is trained to forecast $O_3$ concentrations at all lead hours (from 1 to 96) or days (from 1 to 4), depending on the time scale used (see Sect. 2.4). The features taken into account include a set of chemical features (raw forecast $O_3$ concentration, $O_3$ concentration observed one day before), meteorological features (2-m temperature, 10-m surface wind speed, normalized 10-m zonal and meridian wind speed components, surface pressure, total cloud cover,

surface net solar radiation, surface solar radiation downwards, downward UV radiation at the surface, boundary layer height, and geopotential at 500 hPa; all forecast by the meteorological model) and time features (day of year, day of week, lead hour). Although the past $O_3$ observed concentration corresponds to recursive information that will not be available for all forecast lead days, we use here the same value for all lead days. The tuning of the GBM models is described in Appendix D.

As for QM, the GBM model is first trained (and tuned) only after 30 days to accumulate enough data, and then retrained every 30 days based on all historical data available.

## 2.4 Time scales of MOS corrections

Current AQ standards are defined according to pollutant-dependent time scales, e.g. daily 8-hour maximum (d8max) concentration in the case of $O_3$. In the literature, MOS corrections are typically applied to hourly concentrations, providing hourly corrected concentrations from which the value at the appropriate time scale can then be computed. Following this approach, for a given MOS method X, corrections in this study are first computed based on hourly time series (hereafter referred to as $X_h$), from which daily 24-hour average ($X_d$), daily 1-hour maximum ($X_{d1max}$) and daily 8-hour maximum ($X_{d8max}$) corrected concentrations are then deduced. In addition, MOS corrections are computed directly on daily 24-hour average ($X_{dd}$, the additional "d" indicating that the MOS method is applied directly on daily rather than hourly time series), daily 1-hour maximum ($X_{dd1max}$) and daily 8-hour maximum ($X_{dd8max}$) time series, respectively. When needed, meteorological features are used at the same time scale. This is done to investigate whether applying the MOS correction directly at the regulatory time scale can help achieving better performance.

## 2.5 Evaluation metrics and skill scores

In this study, $O_3$ forecasts are evaluated using an extended panel of continuous and categorical metrics to provide a comprehensive view of the impact of the different MOS methods on the predictions. Continuous metrics used to evaluate the $O_3$ concentrations include :

- nMB : normalized Mean Bias

- nRMSE : normalized Root Mean Square Error

- PCC : Pearson correlation coefficient

- slope : slope of the predicted-versus-observed $O_3$ mixing ratio, to quantify how well lowest and highest $O_3$ concentrations are predicted

- nMSDB : normalized Mean Standard Deviation Bias, to investigate how well the $O_3$ variability is reproduced by the forecast

Categorical metrics used to evaluate the $O_3$ exceedances beyond certain thresholds include :

- H : Hit rate, to quantify the proportion of observed exceedances that are correctly detected

- F : False alarm rate, to quantify the proportion of observed non-exceedances erroneously forecast as exceedances

- FB : Frequency Bias, to investigate to which extent the forecast is predicting the same number of exceedances as observed (no matter if they are predicted on the correct days)

- SR : Success Ratio, to show how much of the predicted exceedances are indeed observed

- CSI : Critical Success Index, to quantify the proportion of correctly predicted exceedances when discarding all the corrected rejections

- PSS : Peirce Skill Score, to investigate to which extent the forecast is able to separate exceedances from non-exceedances

- AUC : Area Under the ROC Curve, to quantify the probability that the forecast predicts higher $O_3$ concentrations during a situation of exceedance compared to a situation of non-exceedance

The formula of these different metrics can be found in Appendix E. Each of them thus highlights a specific aspect of the performance. Regarding categorical metrics, Jolliffe and Stephenson (2011) gave a detailed explanation of the different metric properties desirable for assessing the quality of a forecasting system (see Table 3.4 in Jolliffe and Stephenson, 2011). In this framework, PSS can be considered as the one of the most interesting metrics for assessing the accuracy of the different RAW and MOS-corrected forecasts, given that it gathers numerous valuable properties: (i) truly equitable (all random and fixed-value forecasting systems are awarded the same score, which provides a single no-skill baseline), (ii) not trivial to hedge (the forecaster cannot cheat on his forecast in order to increase PSS), (iii) base rate independent (PSS only depends on H and F, which makes it invariant to natural variations in climate, which is particularly interesting in the frame of AQ forecast where AQ standards and subsequently the base rate can also change) and (v) bounded (values are comprised within a fixed range). It is worth noting that no perfect metric exists, and PSS (as most other metrics) does not benefit from the properties of non-degeneracy (it tends to meaningless values for rare events).

In addition, results are also discussed in terms of skill scores, using the 1-d persistence (PERS(1)) as the reference forecast. Skill scores aim at measuring the accuracy of a forecast relatively to the accuracy of a chosen reference forecast (e.g. persistence, climatology, random choice). They can be computed as $S(X) = (X - X_{\text{reference}})/(X_{\text{perfect}} - X_{\text{reference}})$ with $X$ the score of the forecast, $X_{\text{reference}}$ the score of the PERS(1) reference forecast and $X_{\text{perfect}}$ the score expected with a perfect forecast. Skill scores indicate if a given forecast has a perfect skill (value of 1), a better skill than the reference forecast (value between 0-1), an equivalent skill than the reference forecast (value of 0) or a worse skill than the reference (value below 0, unbounded). To be converted into skill scores, the aforementioned metrics of interest need to be transformed into scores following the rule "the higher the better" (to constrain the skill score to values below 1). For the different metrics $M$, the corresponding score $X(M)$ is obtained applying the following transformations : $X(M) = -M$ for nRMSE and $X(M) = -|1 - M|$ for slope; no transformation are required for the other metrics (H, F, SR, CSI, PSS and AUC). Note that, as indicated by its name, PSS is already intrinsically defined as a skill score (where the reference corresponds to a climatology or random choice, both giving PSS values tending toward 0), but it does not prevent it to be converted into a skill score related to the persistence forecast.

In order to ensure fair comparisons between observations and all the different forecasts, $O_3$ values at a given hour are discarded when at least one of these different dataset does not have data. Over the 2018-2019 period, the resulting data availability exceeds 94% whatever the time scale considered. Note that about 4% of the data is here missing due to the aforementioned minimum of 30 days (i.e. January 2018) of accumulated historical data requested to start computing the corrected forecasts with some MOS methods.

## 3 Results

We first briefly describe the $O_3$ pollution over the Iberian Peninsula as observed by the monitoring stations and simulated by the CAMS regional ensemble forecast (Sect. 3.1). Then, we investigate the performance of the MOS methods on both continuous (Sect. 3.2) and categorical (Sect. 3.3) $O_3$ forecasts. Different sensitivity tests on the MOS methods are performed in Sect. 3.4. Finally, the impact of the input meteorological data on the MOS methods performance is discussed in Sect. 3.4.6.

### 3.1 Ozone pollution over Iberian Peninsula

The European Union sets different standards regarding $O_3$ pollution, including (1) a target threshold of 60 ppbv for the daily 8-hour maximum, with 25 exceedances per year allowed on average over 3 years, (2) an information threshold of 90 ppbv for the daily 1-hour maximum, and (3) an alert threshold of 120 ppbv for the daily 1-hour maximum. In this study, we focus on the two first thresholds and exclude the last one mainly because exceedances of the alert threshold are extremely rare (only 13 exceedances over 314,005 points, i.e. 0.004%). With such a low frequency of occurrence, such events remain extremely difficult to predict (without predicting too many false alarms).

The mean $O_3$ mixing ratios, as well as the annual number of exceedances, are shown in Fig. 1, for both observations and raw CAMS ensemble forecasts. The time series at the different time scales are shown in Fig. 2. Over the Iberian Peninsula, annual mean $O_3$ mixing ratios range between 10 and 50 ppbv, depending on the type of monitoring station (urban traffic, urban background, rural background), with typically higher levels on the Mediterranean coast compared to the Atlantic one. Over the entire domain and time period, the target (d8max > 60 ppbv) and information (d1max > 90 ppbv) thresholds have been exceeded 13,221 and 274 times, respectively (i.e. 4 and 0.08% of the 314,005 points, respectively). These exceedances are well distributed in time along the 2018-2019 period, with 404/730 days (55%) with at least one station exceeding the target threshold, and 78/730 days (11%) with at least one station exceeding the information threshold. These exceedances are observed over a large part of the peninsula, but with a higher frequency in specific locations, including the surroundings (typically downwind) of the largest cities (e.g. Madrid, Barcelona, Valencia, Lisbon, Porto) and close to industrial areas (e.g. Puertollano, a major industrial hot spot at 200 km south of Madrid).

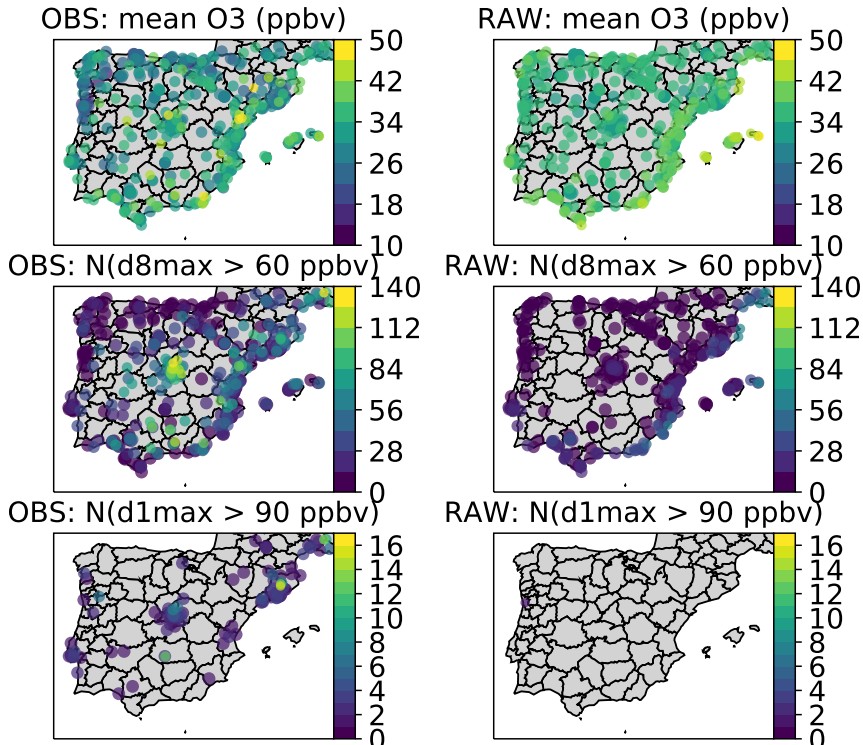

**Figure 1.** Overview of the O$_3$ pollution over the Iberian Peninsula, as observed by monitoring stations (left panels) and as simulated by the CAMS regional ensemble D+1 forecasts (right panels), showing the mean O$_3$ mixing ratios (top panels), and the number of exceedances of the standard (d8max > 60 ppbv; middle panels) and information threshold (d1max > 90 ppbv; bottom panels), over the period 2018-2019. In order to limit the overlap, stations are here plotted by decreasing value and with decreasing size (lowest values with largest symbols but in background, highest values with smallest symbols but in foreground). For clarity, the stations without any observed or simulated exceedance are omitted.

## 3.2 Performance on continuous forecasts

### 3.2.1 RAW forecasts

Considering the annual mean O$_3$ mixing ratios at all 456 stations (Fig. 1), the raw CAMS ensemble forecast represents moderately well the spatial distribution of annual O$_3$ over the Iberian Peninsula (PCC of 0.54 for D+1 forecasts) and strongly underestimates the spatial variability (nMSDB of -42%). At least part of these errors are due to the fact that all station types are taken into account here, including traffic stations where local road transport NOx emissions can strongly reduce the O$_3$ levels (titration by NO), which cannot be properly represented by models at 10 km spatial resolution. In this study, all station types are included because we are ultimately interested in predicting O$_3$ exceedances at all locations where they can be observed

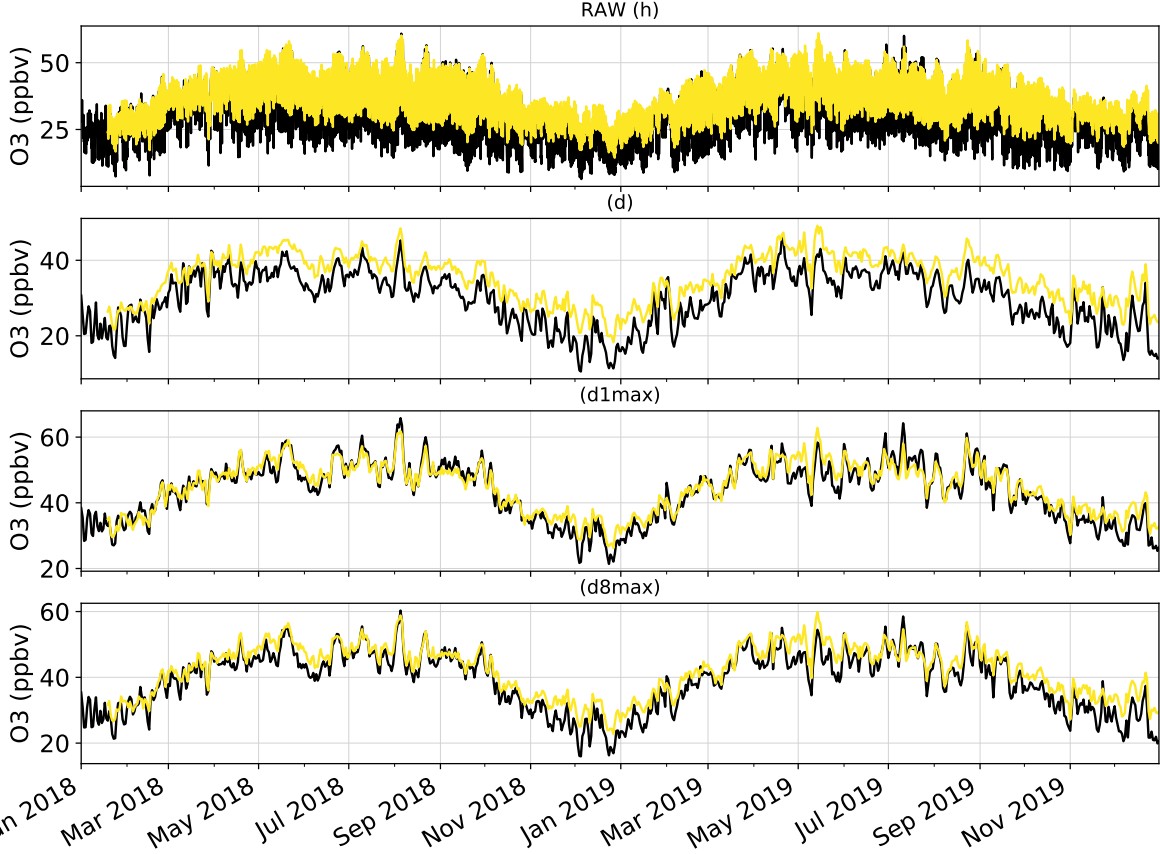

**Figure 2.** Time series of the mean $O_3$ mixing ratios over the Iberian Peninsula, as observed by monitoring stations (in black) and as simulated by the raw CAMS regional ensemble D+1 forecasts (in yellow). Time series are shown at the hourly (h), daily mean (d), daily 1-hour maximum (d1max) and daily 8-hour maximum (d8max) time scales. $O_3$ mixing ratios are averaged over all surface stations of the domain.

(and thus, where air quality standards apply). It is worth noting that the impact of the MOS methods on the different metrics might vary from one type of station to another, although this aspect is beyond the scope of our study. The raw CAMS ensem-
315 ble forecast correctly identifies regions where most exceedances of the target threshold occur but often with underestimated frequency, especially around Madrid, in southern Spain (in-land part of Andalusia region) and along the Mediterranean coast. More severe deficiencies are found with the information threshold that is almost never reached by the CAMS ensemble (with one single exception around Porto).

The overall statistical results are shown in Fig. 3 for the different forecast methods, and a subset of these statistics is given in
Table 1 (and in Table S1 in the Supplement for additional time scales). For a given lead day and time scale, statistics are here computed after aggregating data from all monitoring stations; therefore, statistics of D+1 $O_3$ forecasts at hourly scale can be based on 730 d x 24 h x 455 stations = 7,971,600 points if there are no data gaps. The RAW forecast overestimates moderately

the $O_3$ mixing ratios, especially at hourly and daily time scales, but shows a reasonable correlation at all time scales (above 0.75). However, its main deficiency lies in the underestimated variability (nMSDB around -30%), which is reflected in the low model-versus-observation linear slope obtained (around 0.5-0.6). The deterioration of the performance of the raw CAMS forecasts with lead time is very low, with hourly-scale nRMSE/PCC decreasing from 38%/0.75 at D+1 to 39%/0.72 at D+4, potentially due to their relatively coarse spatial resolution.

As expected (by construction), the PERS(1) reference forecast gives unbiased $O_3$ forecasts. Due to the temporal auto-correlation of $O_3$ concentrations, reasonable results are obtained at D+1 (nRMSE/PCC/slope of 36%/0.74/0.74) but quickly deteriorate with the lead time (down to 42%/0.65/0.64 at D+4). A subset of skill scores with PERS(1) as reference is shown in Fig. 4. Apart from the slope that is always better reproduced by PERS(1), the RAW forecast reaches better skill scores than PERS(1) on both the nRMSE and PCC but only beyond D+1 (with values typically ranging between 0-0.2), and not at all time scales (for instance, PERS(1) systematically shows better RMSE than RAW at daily scale).

### 3.2.2 MOS-corrected forecasts

The MA(1) method removes most of the bias of $O_3$ concentrations and variability. Some residual biases appear when computing the daily 1-h maximum from the MOS-corrected hourly $O_3$ concentrations (i.e. d1max scale), but can be removed by applying the MA(1) method directly at this time scale (i.e. dd1max scale). The MA(1) method substantially improves the other metrics for all lead days, with hourly-scale nRMSE/PCC/slope of 31%/0.81/0.82 at D+1 and 36%/0.74/0.75 at D+4. Thus, the performance still deteriorates with lead time, but slight less dramatically than with PERS(1). In terms of skill scores, such a simple approach as MA(1) is found to strongly improve the skills initially obtained with RAW alone, whatever the time scale or lead time. Skills scores range between 0.1-0.3 for nRMSE and 0.3-0.4 for PCC and slope, with slightly higher values at daily and d8max scales. The variations of skill along lead time differ between nRMSE/PCC (lowest and highest skills typically obtained at D+1 and D+2/D+3/D+4, respectively) and slope (skills tend to progressively decrease from D+1 to D+4, although slightly).

The QM method shows quite similar results than the MA(1) method, but usually with worse (better) performance at short (long) lead time. Thus, the deterioration of the performance with lead time tends to be slower in QM than in MA(1). Biases on $O_3$ concentrations and $O_3$ variability are often slightly higher with QM but remain relatively low (below $\pm5\%$). The strongest improvements of QM compared to MA(1) are found at hourly scale for longest lead times. On these continuous metrics, the skills of the QM method are only slightly positive or even negative at D+1 (except at hourly scale where skill scores are always positive) but are much higher between D+2 and D+4, and often slightly better than MA(1).

Compared to the previous MOS methods, the KF method provides a substantial improvement on both nRMSE and PCC, leading to skill scores of 0.3-0.4 and 0.4-0.6, respectively. However, this comes at the cost of an underestimation of the variability (nMSDB around -10%, still much better than the -30% of nMSDB found in RAW). As for the previous methods, some small biases appear at d1max scale and to a lesser extent at d8max scale but applying this MOS method directly on d1max or d8max $O_3$ mixing ratios rather than hourly data (i.e. dd1max and dd8max scales) mitigates the issue.

Overall, comparable results are found with AN and GBM methods, but the aforementioned issues are typically exacerbated.

**Table 1.** Evaluation of the different forecast methods on continuous metrics, at D+1 (and D+4 into parenthesis), for the h/d/d1max/d8max time scales (see Table S1 in the Supplement for the evaluation results at dd/dd1max/dd8max time scales).

| Time scale | Forecast | nMB | nRMSE | PCC | slope | nMSDB | N |
|---|---|---|---|---|---|---|---|
| h | GBM | -0% (-1%) | 25% (28%) | 0.87 (0.83) | 0.75 (0.71) | -13% (-15%) | 7067085 |
| | AN(10) | 0% (0%) | 26% (28%) | 0.86 (0.82) | 0.75 (0.70) | -13% (-15%) | 7067085 |
| | KF(RMSE) | 0% (-0%) | 25% (28%) | 0.86 (0.83) | 0.78 (0.74) | -10% (-11%) | 7067085 |
| | QM | 3% (3%) | 31% (33%) | 0.81 (0.78) | 0.81 (0.78) | 0% (-1%) | 7067085 |
| | MA(1) | -0% (-1%) | 31% (36%) | 0.81 (0.74) | 0.82 (0.75) | 2% (0%) | 7067085 |
| | PERS(1) | 0% (0%) | 36% (42%) | 0.75 (0.65) | 0.75 (0.65) | 0% (-0%) | 7067085 |
| | RAW | 18% (17%) | 38% (39%) | 0.75 (0.72) | 0.53 (0.50) | -29% (-30%) | 7067085 |
| d | GBM | -1% (-1%) | 16% (18%) | 0.91 (0.88) | 0.84 (0.80) | -7% (-9%) | 295617 |
| | AN(10) | 0% (0%) | 16% (19%) | 0.90 (0.86) | 0.78 (0.73) | -13% (-15%) | 295617 |
| | KF(RMSE) | 0% (-0%) | 15% (18%) | 0.91 (0.88) | 0.85 (0.80) | -7% (-9%) | 295617 |
| | QM | 3% (2%) | 20% (22%) | 0.86 (0.84) | 0.91 (0.87) | 5% (4%) | 295617 |
| | MA(1) | -0% (-1%) | 16% (22%) | 0.91 (0.82) | 0.92 (0.81) | 1% (-2%) | 295617 |
| | PERS(1) | 0% (0%) | 20% (29%) | 0.85 (0.70) | 0.85 (0.70) | -0% (-0%) | 295617 |
| | RAW | 18% (17%) | 30% (30%) | 0.76 (0.74) | 0.55 (0.52) | -28% (-29%) | 295617 |
| d1max | GBM | -8% (-8%) | 16% (18%) | 0.86 (0.83) | 0.80 (0.75) | -8% (-10%) | 295617 |
| | AN(10) | -4% (-4%) | 15% (17%) | 0.86 (0.82) | 0.74 (0.70) | -14% (-15%) | 295617 |
| | KF(RMSE) | -3% (-4%) | 13% (15%) | 0.89 (0.85) | 0.81 (0.77) | -8% (-10%) | 295617 |
| | QM | -1% (-1%) | 17% (18%) | 0.82 (0.80) | 0.83 (0.80) | 1% (-0%) | 295617 |
| | MA(1) | 3% (2%) | 15% (18%) | 0.86 (0.79) | 0.87 (0.77) | 1% (-2%) | 295617 |
| | PERS(1) | 0% (0%) | 17% (23%) | 0.82 (0.67) | 0.82 (0.67) | -0% (-1%) | 295617 |
| | RAW | 2% (2%) | 19% (19%) | 0.76 (0.74) | 0.55 (0.52) | -28% (-29%) | 295617 |
| d8max | GBM | -4% (-5%) | 15% (17%) | 0.89 (0.86) | 0.83 (0.79) | -7% (-8%) | 295617 |
| | AN(10) | -1% (-2%) | 15% (17%) | 0.88 (0.85) | 0.78 (0.73) | -12% (-14%) | 295617 |
| | KF(RMSE) | -1% (-2%) | 13% (15%) | 0.91 (0.88) | 0.85 (0.81) | -7% (-8%) | 295617 |
| | QM | 1% (2%) | 17% (19%) | 0.85 (0.83) | 0.88 (0.84) | 3% (1%) | 295617 |
| | MA(1) | 1% (0%) | 15% (18%) | 0.89 (0.83) | 0.89 (0.81) | 0% (-2%) | 295617 |
| | PERS(1) | 0% (0%) | 18% (24%) | 0.84 (0.70) | 0.84 (0.70) | -0% (-1%) | 295617 |
| | RAW | 7% (7%) | 21% (22%) | 0.79 (0.76) | 0.57 (0.54) | -27% (-29%) | 295617 |

The negative biases at d1max and d8max time scales are much higher, especially for GBM, but can be removed at dd1max and dd8max scales. Similarly, the underestimation of the variability is much more pronounced, with nMSDB values around -15% and -10% for AN and GBM, respectively. These two MOS methods thus show a good performance for predicting the central part of the distribution of $O_3$ mixing ratios, but have more difficulty in capturing the lowest and highest $O_3$ concentrations observed on the tails of this distribution. Besides the negative nMSDB, this typically leads to lower slopes compared to the other MOS methods. Skill scores on nRMSE and PCC span over a relatively large range of values depending on the time scale and the lead time. They are typically the lowest at short lead times and/or at specific time scales (e.g. d1max) but can reach among the highest values (although slightly lower than KF), for instance with GBM, at hourly and daily scale at D+2/D+3/D+4. Concerning the slope, the aforementioned issues are here illustrated by the typically low skills of both AN and (to a slightly lesser extent) GBM methods, often worse than the other MOS methods.

Therefore, on this set of continuous metrics, the impact of the MOS corrections on the performance strongly varies with the method considered. Among the different MOS methods, KF seems to give the most balanced improvement with biases mostly removed, errors and correlation substantially improved and variability not too strongly underestimated. However, it is worth noting that since some MOS methods (namely QM, AN and GBM) can ingest increasing amounts of input data over time, we can expected their performance to change (increase) between the beginning of the period when very limited past data information is available and the end of the period when more past data have been accumulated. Investigating this aspect would ideally require a proper analysis, comparing the performance obtained over a given period using variable amount of past input data. Here, we simply provide some insights by comparing the relative difference of performance of these MOS methods against RAW, (1) when evaluated over the entire 2018-2019 period (i.e. including the beginning of the period of study when MOS methods can only rely on limited past data), and (2) when evaluated only over the year 2019 (i.e. when the first year is discarded). In the first case (evaluation over 2018-2019), the QM, AN and GBM show nRMSE 31, 41 and 44% lower than RAW, respectively. In the second case (evaluation over 2019), these MOS methods give nRMSE 33, 44 and 49% lower than RAW. Therefore, this basic comparison suggests that these MOS methods can indeed benefit from a larger amount of past data. Here, the change is more pronounced more GBM, which suggests that this MOS method is the one benefiting the most from more past training data. For GBM, this improvement is mainly due to the relatively poor predictions made during the very first months of 2018 when the training dataset was the most limited (see time series in Fig. F1 in Appendix F).

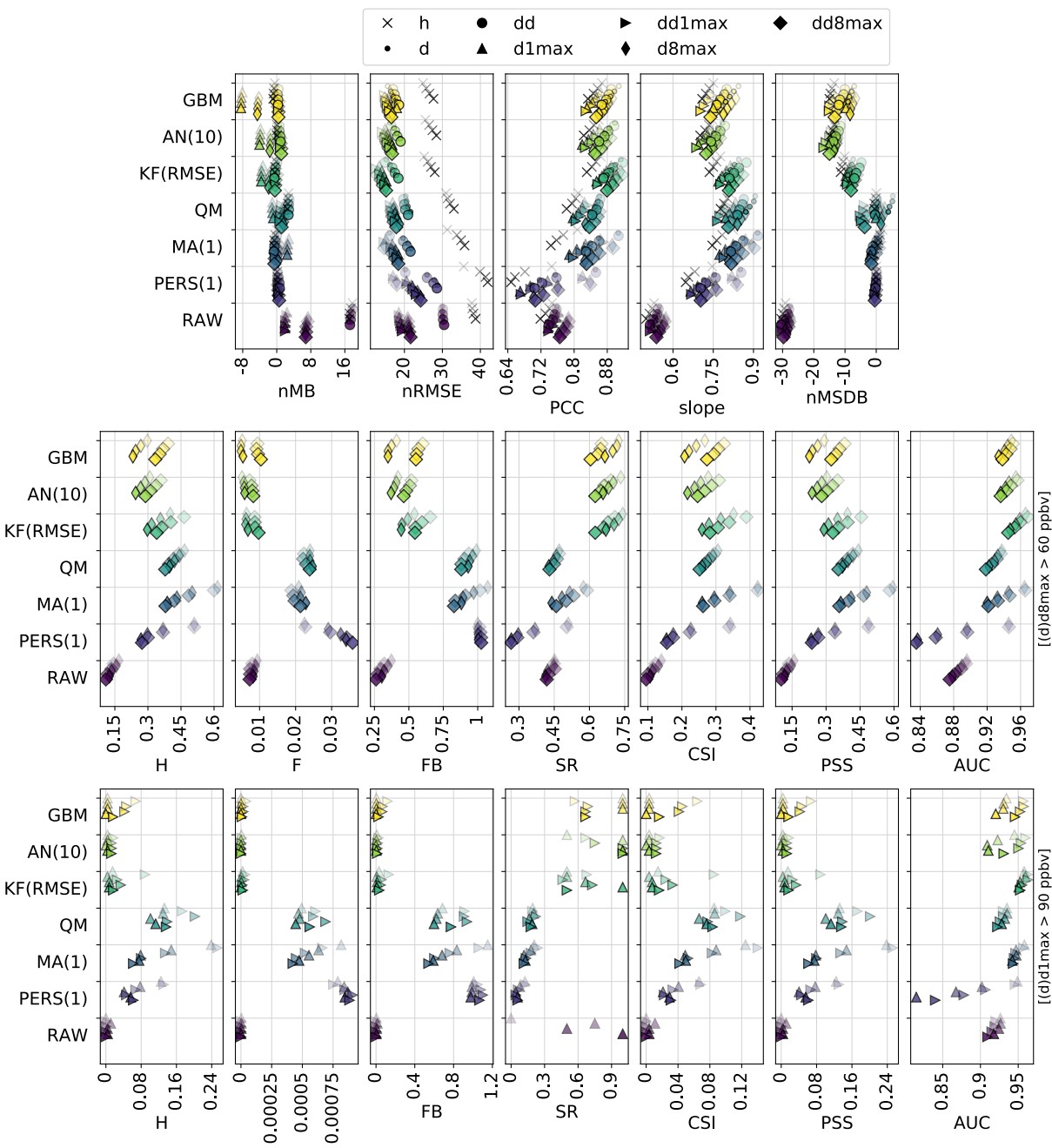

**Figure 3.** Statistical performance of RAW and MOS-corrected CAMS $O_3$ forecasts for continuous metrics (top panels) and categorical metrics related to the exceedance of the target (intermediate panels) and information threshold (bottom panels). The different symbols depict results obtained at different time scales (h: hourly; d: daily mean; d1max/dd1max: daily 1-hour maximum; d8max/dd8max: daily 8-hour maximum). In each panel, results are shown for the different methods (each with a given color). The overlaying symbols of decreasing transparency show the results at the different lead days from D+1 (most transparent) to D+4 (most opaque). Metrics : normalized Mean Bias (nMB in %), normalized Root Mean Square Error (nRMSE in %), Pearson correlation coefficient (PCC), slope (unitless), normalized Mean Standard Deviation bias (nMSDB in %), Hit rate (H), False alarm rate (F), Frequency Bias (FB), Success Ratio (SR), Critical Success Index (CSI), Peirce Skill Score (PSS), Area Under the ROC Curve (AUC). See Sect. 2.4 and 2.5 for details on time scales and metrics, respectively.

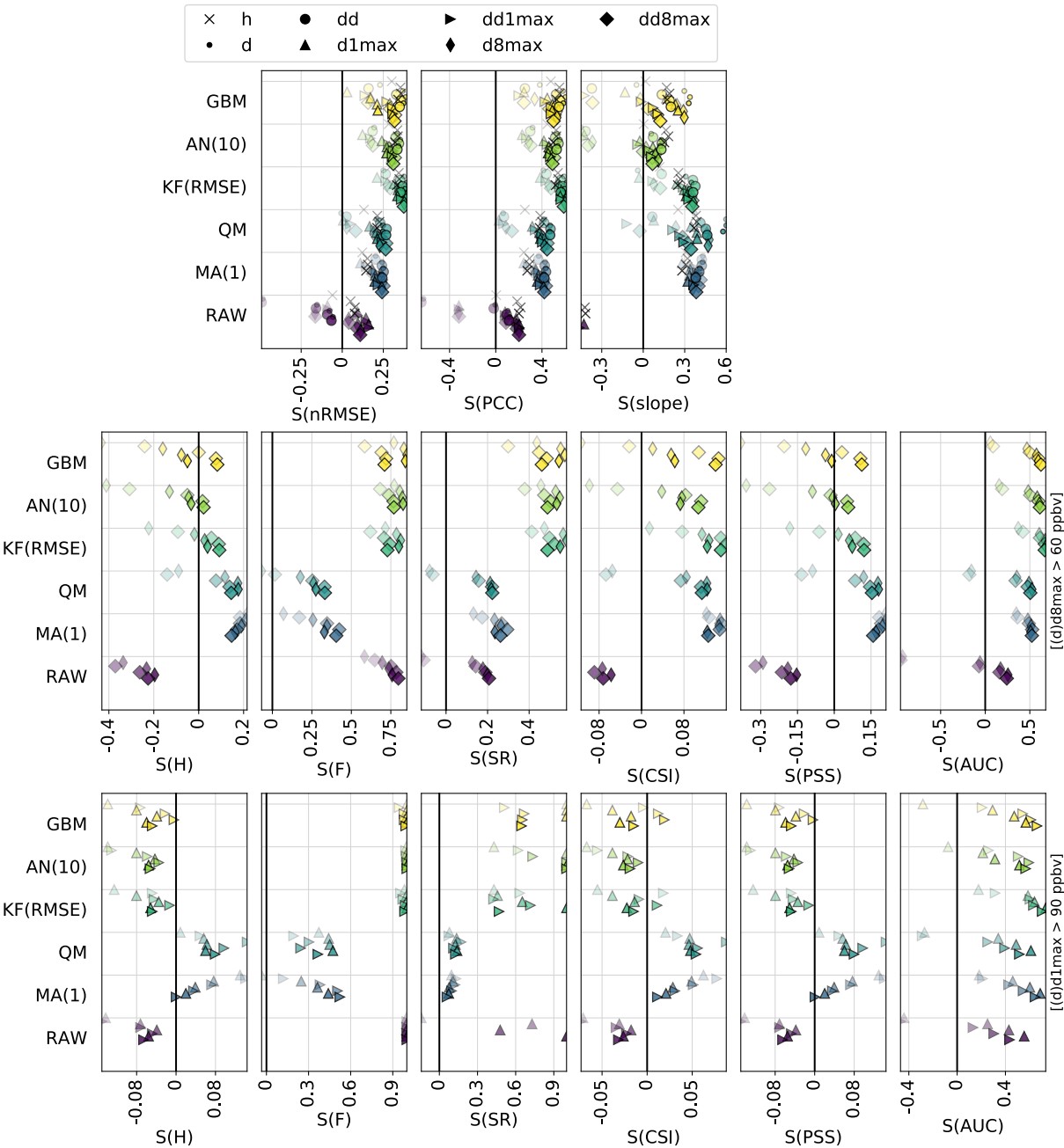

**Figure 4.** Similar to Fig. 3 for skill scores (see Sect. 2.5 for details on the calculation of these skill scores). For clarity, highest negative values (mostly obtained on RAW and/or shortest lead times) are cut but can be seen in Fig. S1 in the Supplement.

### 3.3 Performance on categorical forecasts

**3.3.1 RAW forecasts**

Focusing now on the performance for detecting target and information thresholds, Fig. 3 (middle and bottom panels) shows a comprehensive set of metrics, where the most interesting ones are probably CSI and PSS, followed by SR and AUC.

The RAW forecast shows low H and F (very few true positives and false negatives). With an intermediate SR (0.45, i.e. only 45% of the exceedances predicted by RAW indeed occur), it can be seen as a moderately "conservative" forecast for target

thresholds (d8max $O_3$ above 60 ppbv); the term "conservative" here refers to forecasting systems that predict exceedances only with strong evidence (it thus predicts very few exceedances but with a moderate confidence). Despite showing a reasonably good AUC, the RAW forecast strongly fails at reproducing high $O_3$ mixing ratios, as illustrated by the low FB (0.25, i.e. RAW predicts 4 times less exceedances than the observations), and finally shows the worst performance in terms of CSI (0.10) or PSS (0.15). In comparison, the PERS(1) reference forecast provides better detection skills regarding target thresholds. This is

especially true at short lead days, but the performance then quickly decreases with the lead time, with CSI/PSS reduced from about 0.27/0.42 at D+1 to about 0.14/0.23 at D+4. Except FB, all categorical metrics show a similarly strong sensitivity to the lead time. With PERS(1) taken as a reference, the skill scores of RAW clearly show negative and positive values for H and F, respectively (i.e. it predicts less true exceedances but produces less false alarms). The consequence in terms of SR skills is positive but only beyond D+1. With positive skills on AUC, RAW is able to discriminate exceedances and non-exceedances

slightly better than PERS(1), but only beyond D+2. However, its skills on the important CSI and PSS metrics are strongly negative at all lead times, which highlights its overall deficiency for predicting correctly the exceedances of the target threshold (i.e. without too many false alarms).

Exceedances of the information threshold (d1max $O_3$ above 90 ppbv) appear even more difficult to capture for the RAW forecast with CSI and PSS typically below 0.02. However, given that it is also more difficult for PERS(1) to capture these

405 exceedances, the skills of RAW on these two metrics are substantially better (although still negative) on this information threshold compared to the target threshold. Results also show much better SR, especially at longest lead times (i.e. most of the predicted exceedances indeed occur), but this apparently good result has to be put in front of the extremely low H (i.e. RAW almost never predict exceedances).

**3.3.2 MOS-corrected forecasts**

Although the RAW forecast alone shows quite limited skills for predicting high $O_3$ exceedances, its potential usefulness is nicely illustrated by the results obtained when it is combined with observations, such as in MA(1), QM or KF(RMSE). When considering the target threshold exceedances, CSI and PSS are indeed greatly improved with these last MOS methods, and to a lesser extent by the two other methods, AN(10) and GBM. KF(RMSE), AN(10) and GBM clearly appear as the most "conservative" MOS approaches here, with relatively low H and F, but strong SR. In other terms, they predict fewer

exceedances but with a higher reliability. In terms of skill scores, all these MOS-corrected forecasts always have better skills than RAW. However, only MA(1) always beats PERS(1) at all lead times, while the other MOS methods provide positive

skills only beyond D+1/D+2. This MA(1) method thus clearly outperforms the other methods at D+1, while differences of performance are reduced when considering longer lead times. At longer lead times, the ranking between these different MOS methods varies substantially depending on the considered metric, with MA(1), KF(RMSE) and GBM showing best skills on CSI, and MA(1) and QM showing best skills on PSS.

However, when considering the detection of the information threshold, the KF(RMSE), AN(10) and GBM methods still benefit from a strong SR but are missing too many of the observed exceedances, which leads to a dramatic deterioration of both CSI and PSS. As for RAW, this means that there is a high change that an exceedance predicted by these methods indeed occurs but such exceedances are too rarely predicted. Most of their skill scores on PSI are found to be negative, while only a few positive skills are obtained on CSI for specific time scales in KF and GBM methods. For detecting such high $O_3$ values, best methods are finally MA(1) for shortest lead times. At longer lead times, the skills of MA(1) quickly deteriorate and best skills are finally obtained for QM. Both methods reproduce fairly well the geographical distribution of high $O_3$ episodes (PERS(1) reproduces it perfectly, by construction), as shown in Fig. 5, but still with very low SR (below 0.25 for exceedances of the information threshold).

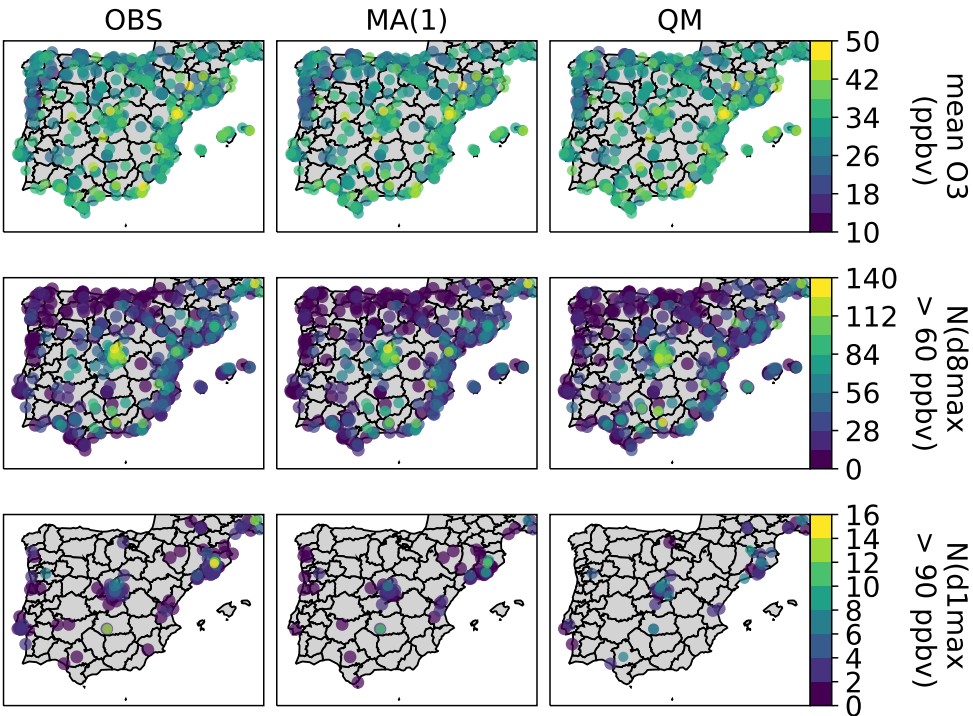

**Figure 5.** Similar to Fig. 1 but for observations, and D+4 $O_3$ forecasts corrected with MA(1) and QM methods.

**Table 2.** Evaluation of the different forecast methods on categorical metrics, at D+1 (and D+4 into parenthesis), for both target and information thresholds.

| Time scale and threshold | Forecast | H | F | SR | CSI | PSS | AUC | N |
|---|---|---|---|---|---|---|---|---|
| d8max>60 | GBM | 0.30 (0.23) | 0.01 (0.01) | 0.72 (0.67) | 0.27 (0.21) | 0.29 (0.23) | 0.95 (0.93) | 295617 |
|  | AN(10) | 0.31 (0.24) | 0.01 (0.01) | 0.73 (0.66) | 0.28 (0.22) | 0.30 (0.24) | 0.95 (0.94) | 295617 |
|  | KF(RMSE) | 0.40 (0.30) | 0.01 (0.01) | 0.74 (0.67) | 0.35 (0.26) | 0.39 (0.29) | 0.97 (0.95) | 295617 |
|  | QM | 0.47 (0.40) | 0.02 (0.02) | 0.47 (0.43) | 0.31 (0.26) | 0.44 (0.37) | 0.94 (0.92) | 295617 |
|  | MA(1) | 0.62 (0.39) | 0.02 (0.02) | 0.57 (0.44) | 0.42 (0.26) | 0.59 (0.36) | 0.96 (0.92) | 295617 |
|  | PERS(1) | 0.51 (0.27) | 0.02 (0.03) | 0.51 (0.27) | 0.34 (0.15) | 0.49 (0.23) | 0.95 (0.84) | 295617 |
|  | RAW | 0.17 (0.13) | 0.01 (0.01) | 0.45 (0.41) | 0.14 (0.11) | 0.16 (0.12) | 0.90 (0.88) | 295617 |
| dd8max>60 | GBM | 0.39 (0.33) | 0.01 (0.01) | 0.65 (0.60) | 0.32 (0.27) | 0.38 (0.32) | 0.95 (0.94) | 286803 |
|  | AN(10) | 0.36 (0.29) | 0.01 (0.01) | 0.69 (0.62) | 0.31 (0.25) | 0.35 (0.28) | 0.96 (0.94) | 286803 |
|  | KF(RMSE) | 0.46 (0.34) | 0.01 (0.01) | 0.71 (0.62) | 0.39 (0.28) | 0.46 (0.33) | 0.97 (0.95) | 286803 |
|  | QM | 0.44 (0.38) | 0.02 (0.02) | 0.47 (0.43) | 0.29 (0.25) | 0.42 (0.35) | 0.94 (0.92) | 286803 |
|  | MA(1) | 0.60 (0.38) | 0.02 (0.02) | 0.59 (0.46) | 0.42 (0.26) | 0.58 (0.36) | 0.97 (0.92) | 286803 |
|  | PERS(1) | 0.51 (0.27) | 0.02 (0.04) | 0.50 (0.27) | 0.34 (0.16) | 0.49 (0.24) | 0.95 (0.84) | 286803 |
|  | RAW | 0.14 (0.11) | 0.01 (0.01) | 0.45 (0.42) | 0.12 (0.09) | 0.14 (0.10) | 0.89 (0.88) | 286803 |
| d1max>90 | GBM | 0.00 (0.00) | 0.00 (0.00) | 1.00 (nan) | 0.00 (0.00) | 0.00 (0.00) | 0.93 (0.92) | 295617 |
|  | AN(10) | 0.00 (0.00) | 0.00 (0.00) | 0.50 (1.00) | 0.00 (0.00) | 0.00 (0.00) | 0.95 (0.91) | 295617 |
|  | KF(RMSE) | 0.02 (0.01) | 0.00 (0.00) | 0.50 (1.00) | 0.01 (0.01) | 0.02 (0.01) | 0.96 (0.95) | 295617 |
|  | QM | 0.13 (0.11) | 0.00 (0.00) | 0.19 (0.19) | 0.09 (0.08) | 0.13 (0.11) | 0.94 (0.93) | 295617 |
|  | MA(1) | 0.24 (0.08) | 0.00 (0.00) | 0.21 (0.13) | 0.12 (0.05) | 0.24 (0.07) | 0.96 (0.94) | 295617 |
|  | PERS(1) | 0.12 (0.06) | 0.00 (0.00) | 0.12 (0.06) | 0.07 (0.03) | 0.12 (0.06) | 0.95 (0.82) | 295617 |
|  | RAW | 0.00 (0.00) | 0.00 (0.00) | 0.00 (1.00) | 0.00 (0.00) | -0.00 (0.00) | 0.93 (0.92) | 295617 |
| dd1max>90 | GBM | 0.07 (0.02) | 0.00 (0.00) | 0.57 (0.67) | 0.06 (0.02) | 0.07 (0.02) | 0.96 (0.95) | 288980 |
|  | AN(10) | 0.02 (0.01) | 0.00 (0.00) | 0.67 (1.00) | 0.02 (0.01) | 0.02 (0.01) | 0.96 (0.93) | 288980 |
|  | KF(RMSE) | 0.09 (0.02) | 0.00 (0.00) | 0.68 (0.50) | 0.09 (0.02) | 0.09 (0.02) | 0.96 (0.95) | 288980 |
|  | QM | 0.17 (0.14) | 0.00 (0.00) | 0.19 (0.18) | 0.10 (0.08) | 0.17 (0.14) | 0.93 (0.92) | 288980 |
|  | MA(1) | 0.25 (0.06) | 0.00 (0.00) | 0.24 (0.11) | 0.14 (0.04) | 0.25 (0.06) | 0.96 (0.94) | 288980 |
|  | PERS(1) | 0.13 (0.06) | 0.00 (0.00) | 0.12 (0.06) | 0.07 (0.03) | 0.13 (0.06) | 0.95 (0.84) | 288980 |
|  | RAW | 0.00 (0.00) | 0.00 (0.00) | nan (nan) | 0.00 (0.00) | 0.00 (0.00) | 0.92 (0.91) | 288980 |

## 3.4 Sensitivity tests

Each of the forecast methods considered in this study relies on a specific configuration, e.g. the time window of PERS or MA methods, the metric used internally in KF for optimizing the variance ratio, the number of analogs taken into account in AN, the choice of input features or metrics used internally for fitting the ML model in GBM. This configuration can substantially influence their general performance, although in a different way depending on the metric used. In the previous sections, we evaluated the performance of these different methods considering a relatively simple baseline configuration. In this section, we discuss some of these choices and investigate their impact on the performance through different sensitivity tests. Corresponding statistical results on continuous and categorical metrics are given in Tables in the Supplement.

### 3.4.1 Persistence method

The persistence method with a 1-d time window (PERS(1)) provides a reference forecast for assessing the skill scores on the different RAW and MOS-corrected forecasts. Here we explore how the time window, from 1 to 10 d (hereafter referred to as PERS($n$) with $n$ the window in days), impacts the performance of this PERS forecasts. Results are shown in Fig. G1 in Appendix G.

Increasing the window leads to a growing negative bias on d1max and d8max scales that can be substantially reduced when working at dd1max and dd8max scales, i.e. when applying the PERS approach directly on daily 1-hour and 8-hour maxima rather than on the hourly time series. The differences between the two approaches originate from the day-to-day variability in the hour of the day when $O_3$ mixing ratios peak. For illustration purposes, let's assume that $O_3$ peaks between 15 and 17 h; on a given day, $O_3$ mixing ratios at 15/16/17h reach 50/60/50 ppbv and on the following day 70/70/80 ppbv. Then, the PERS(2)$_{dd1max}$ $O_3$ would be 70 ppbv (mean of 60 and 80 ppbv), while the PERS(2)$_{d1max}$ $O_3$ would be only 65 ppbv (maximum of the mean diurnal profile of these two days, in this case 60/65/65). Conversely, both nRMSE and PCC can be slightly improved with longer windows, but at the cost of a growing underestimation of the variability. As a consequence, both H and F are slightly reduced, which means that PERS forecasts become more "conservative" with longer windows. The impact on SR for detecting exceedances of the target threshold is low for short lead times but positive for the longest ones. Interestingly, for information thresholds, the best SR are obtained around 4-7 d. However and more importantly, using longer windows deteriorates the general performance of the forecast, as shown by the decrease of both CSI and PSS, especially at short lead times. Interestingly, there are also important differences in terms of AUC for detecting exceedances of the target threshold depending on the lead day, ranging from a decrease of AUC with longer windows at D+1 to an increase at D+4. Therefore, for detecting exceedances, considering PSS and/or CSI as the most relevant metrics, the PERS method shows its best performance for a time window of 1 d. However, it gives very "liberal" $O_3$ forecasts with rather poor SR. The term "liberal" is here borrowed from (Fawcett, 2006) to designate forecasting systems that predict exceedances with weak evidence, in opposition with the aforementioned term "conservative". Longer time windows can improve SR, but result in an important deterioration of CSI and PSS, particularly for the shorter lead times (D+1/D+2).

### 3.4.2 Moving average method

Here, a sensitivity test is performed on MA with windows ranging between 1 and 10 d (hereafter referred to as MA($n$) with $n$ the window in days). Results are shown in Fig. G2 in Appendix G. Increasing the window length impacts the MA performance in a very similar way than for PERS, especially for continuous metrics. Regarding the detection of the target threshold, the main noticeable difference is the absence of strong deterioration of some metrics like AUC, SR or CSI for shorter lead times. Regarding the detection of the information threshold, the clearest difference with PERS concerns the SR that substantially improves when considering longer windows. However, the deterioration of both CSI and PSS persists.

Therefore, the detection of $O_3$ exceedances with the MA method shows its best performance with shortest windows (1 d). As for PERS, the corresponding forecasts are quite liberal with low SR. However, in contrast to PERS, the SR associated to high thresholds can be substantially improved when using longer windows, which may be an interesting option if the corresponding deterioration of CSI/PSS is seen as acceptable.

### 3.4.3 Kalman filter method

As explained in Sect. 2.3.3 (and Appendix B), the behavior of the KF intrinsically depends on the $\sigma_\eta^2/\sigma_\epsilon^2$ ratio chosen. So far, this parameter has been adjusted dynamically (and updated regularly) to optimize the RMSE on past data. Here, a sensitivity test is performed with alternative strategies in which the variance ratio is chosen to optimize the SR, CSI, PSS or AUC with threshold values of 60 or 90 ppbv (hereafter referred to as SR-60, SR-90, CSI-60, CSI-90, PSS-60, PSS-90, AUC-60 and AUC-90). The objective is to investigate to what extent tuning the KF algorithm with appropriate categorical metrics allows improving the exceedance detection skills.

Results (Fig. G3 in Appendix G) show that this tuning strategy barely impacts the performance obtained on continuous metrics, except for CSI-60 and PSS-60 that show slightly deteriorated RMSE and PCC. Only small differences are also found on target threshold exceedances, except again with these two methods that show slightly improved CSI/PSS at short lead time. Results on information threshold exceedances show more variability depending on the time scale, but both CSI and PSS can typically be improved when used internally in the KF procedure, although often only at short lead times. The choice of the threshold in this optimizing metric leads to more ambiguous results. For instance, besides giving the best PSS on target threshold, KF(PSS-60) also gives better results than KF(PSS-90) on the information threshold. Reasons behind this behavior are not clear but may be due to some instabilities brought into PSS-90 by the rareness of such exceedances. Indeed, a common and well-known issue of PSS (as well as CSI and most other categorical metrics) is that it degenerates to trivial values (either 0 or 1) for rare events : as the frequency of the event decreases, the numbers of hits (a), false alarm (b) and missed exceedances (c) all decay toward zero but typically at different rates, which causes the metric to take meaningless values (either 0 or 1 in the case of PSS) (Jolliffe and Stephenson, 2011; Ferro and Stephenson, 2011). All in all, the performance for detecting such high $O_3$ concentrations remains very poor, especially far in time, but this sensitivity test demonstrates that choosing an appropriate tuning strategy can help improving slightly the detection skills at a potential cost in terms of continuous metrics.

### 3.4.4 Analog method

The AN method identifies the closest analog days to estimate the corresponding prediction, and thus depends on the number of analog days taken into account. We performed a sensitivity test with 1, 5, 10, 15, 20, 25 and 30 analog days (hereafter referred to as AN(N) with N the number of analogs). Results are shown in Fig. G4 in the Appendix G.

Although the best slopes are found with smallest number of analogs, the best nRMSE and PCC are obtained using around 5-15 analogs. Using too numerous analogs increases the underestimation of the variability and deteriorates the slope. Regarding the

500 detection of target thresholds, increasing the number of analogs makes the forecast more "conservative" (lower H and F, higher SR) and deteriorates the CSI and PSS. When focusing on information threshold exceedances, the AN forecasts based on 10 analogs or more never reach such high $O_3$ values. Highest CSI and PSS are finally obtained with one single analog.

Therefore, similarly to PERS and MA methods that reached their best skills for the shortest time windows, with AN the best CSI and PSS skills are obtained when using the lowest number of analogs (with a cost in the continuous metrics, as for PERS

and MA). Computing the AN-corrected $O_3$ mixing ratios based on a larger number of analogs gives smoother predictions, and our choice to weight the average by the distance to the different analogs is unable to substantially mitigate this issue.

### 3.4.5 Gradient boosting machine method

Although GBM gives among the best RMSE and PCC, it strongly underestimates the variability of $O_3$ mixing ratios, with critical consequences in terms of detection skills, especially for the highest thresholds (e.g. d1max > 90 ppbv). This is at least

510 partly due to the low frequency of occurrence of such episodes, and their corresponding low weight in the entire population of points used for the training. One way of mitigating this issue consists in specifying different weights to the different training instances. This aims at forcing the GBM model to better predict the instances of higher weight, at the cost of a potential deterioration of the performance on the instances of lower weight.

In order to assess to which extent it may improve the performance of the GBM MOS method, we here test different weighting

strategies. At each training phase, we compute the absolute distance $D$ between all observed $O_3$ mixing ratio instances and the mean $O_3$ mixing ratio (averaged over the entire training dataset). Then several sensitivity tests are performed, weighting the training data by $D$, $D^2$ and $D^3$, respectively (hereafter referred to as GBM(W), GBM(W2), GBM(W3), respectively). Using such weights, we want the GBM model to better predict the lower and upper tails of the $O_3$ distribution in order to better represent the variability of the $O_3$ mixing ratios. Given that the $O_3$ mixing ratio distribution is typically positively skewed, the

highest weights are put on the strongest positive deviations from the mean.

As a parallel sensitivity test, we explore the performance of these different ML models but removing the input feature corresponding to the previous (one day before) observed $O_3$ mixing ratio (hereafter referred to as GBM(noO), GBM(noO,W), GBM(noO,W2) and GBM(noO,W3)). This additional test is of interest for operational purposes since $O_3$ observations are not always available in near real-time. Results are shown in Fig. G5 in the Appendix G.

As expected, results highlight a deterioration of the RMSE and PCC combined with an improvement of the slope and nMSDB. The negative bias affecting the variability with the unweighted GBM is substantially reduced when using weights, although

too strong weights (as in GBM(W3) for instance) can lead to a slight overestimation of the variability at specific time scales. Regarding the skills for detecting target threshold exceedances, stronger weights typically increase both H and F, improve the (underestimated) FB, but deteriorate the SR and AUC (the forecasts become more liberal). Regarding the more balanced metrics (of strongest interest here), adding more weights on the tails of the $O_3$ distribution typically has a positive although small impact on CSI and PSS. Regarding the detection of information threshold exceedances, both CSI and PSS can also be slightly improved by adding some weight into the GBM, but the performance for detecting such high $O_3$ values remain relatively low. The interest of using the $O_3$ concentration observed one day before is here found to be limited.

Therefore, adopting an appropriate weighting strategy is simple yet effective for achieving slightly better $O_3$ exceedance detection skills in exchange of a reasonable deterioration in RMSE and PCC. Overall, the improvements are relatively small, but still valuable given the initially very low detection skills for the strongest $O_3$ episodes.

### 3.4.6 Influence of the meteorological input data in AN and GBM methods

In the previous sections, $O_3$ corrections with AN and GBM methods relied on HRES meteorological forecasts. Here, we investigate the impact of using an alternative meteorological data, namely the ERA5 meteorological reanalysis. For both AN and GBM methods, the MOS-corrected $O_3$ mixing ratios obtained with these two meteorological dataset are very similar, with PCC above 0.95. The results obtained against observations are shown in Fig. G6 in the Appendix G, for the AN(1), AN(5), AN(10) and GBM methods. Since $O_3$ predictions are close, the statistical performance against observations is also very consistent between both meteorological datasets. For both continuous and categorical metrics, the performance obtained with HRES data is found to be slightly lower than with ERA5. Discrepancies between both meteorological dataset tend to increase with lead time, with GBM being slightly more sensitive to the meteorological input data than AN.

Therefore, this experiment highlights a relatively low sensitivity of both AN and GBM methods to the two meteorological datasets tested here. The very similar results obtained with IFS and ERA5 meteorological input data are likely not explained by the fact that both datasets give very similar values for the different meteorological variables, but rather by the intrinsic characteristics of both AN and GBM methods. The AN method make use of the meteorological data only to identify past days with more or less similar meteorological conditions, and can thus handle to some extent the presence of biases in meteorological variables as far as they are systematic (and thus do not impact the identification of the analogs). On the other side, the GBM method uses past information to learn the complex relationship between $O_3$ mixing ratios and the other ancillary features. Although the better the input data, the higher the chances are to fit a reliable model for predicting $O_3$, the GBM models can also learn indirectly at least part of the potential errors affecting some meteorological variables and how they relate to $O_3$ mixing ratios. Therefore, the presence of biases in some of the ancillary features is not expected to strongly impact the performance of the predictions.

## 4 Discussion and conclusions

We demonstrated the strong impact of MOS methods to enhance raw CAMS $O_3$ forecasts, not only by removing potential systematic biases but also for correcting other issues related to the distribution and/or variability of $O_3$ mixing ratios. All MOS approaches were indeed able to substantially improve at least some aspects of the RAW $O_3$ forecasts, first and foremost the RMSE and PCC, for which the strongest improvements are obtained with most sophisticated MOS methods like KF, AN or GBM. However, although all MOS methods were able to increase the underestimated variability of $O_3$ mixing ratios of RAW, the strongest improvements of slope and nMSDB were obtained with more simple MOS methods like MA or QM. $O_3$ mixing ratios corrected with AN, GBM and to a lesser extent KF, remained too smooth, and such a deficiency has a major impact on the detection skills for high $O_3$ thresholds. All in all, the best PSS or CSI are usually obtained with the more simple MOS methods. Therefore, there is a clear trade-off between the continuous and categorical skills scores, as also shown by the different sensitivity tests. The quality of a MOS-corrected forecast assessed solely based on metrics like RMSE or PCC thus tells little about the forecast value, here understood as an information a user can benefit from to make better decisions, notably for mitigating $O_3$ short-term episodes.

More generally, our study highlights the complexity of identifying the "best" MOS method given the multiple dimensions of the problem. The relative performance of the MOS methods can vary depending on the metric used, the threshold considered in the case of categorical metrics (or more specifically the base rate), the time scale at which MOS corrections are computed and/or evaluated, or the lead time. Other dimensions not covered by this study, like the seasonality of the performance, are also susceptible of shedding a different light on the inter-comparison.

Among the continuous metrics, both RMSE and PCC provide a first valuable information on the performance of a MOS method. However, a MOS method can give the best RMSE and PCC, yet the poorest high $O_3$ detection skills. This was the case of the unweighted GBM method. Continuous metrics like the model-versus-observation linear slope or nMSDB provide important complementary information, potentially less misleading, especially in a context where the final objective is to predict episodes of strong $O_3$. Among the categorical metrics, although results were presented on a relatively large set of metrics, all metrics do not benefit from the same properties. PSS may be considered as one of the most valuable, notably due to its independence from the base rate, in contrast to CSI. Such a property is particularly useful when comparing scores over different regions and/or time periods where the frequency of observed exceedances might vary, for instance due to different emission forcing and/or meteorological conditions. In an operational context where statistical metrics are continuously monitored, the independence from the base rate is an interesting property because it may change with time, which prevents from a consistent comparison between different periods. However, a well-known issue of both PSS and CSI (as well as many other categorical metrics) is that they degenerate to trivial values (either 0 or 1) as events become rarer (Jolliffe and Stephenson, 2011; Ferro and Stephenson, 2011), which should restrict their use to the detection of not too rare (and therefore not too high) $O_3$ episodes. In this study, the base rate of the target threshold was likely sufficiently high (s around 5%), but we were probably already at the limit regarding the information threshold (s around 0.1%). All in all, the selection of the evaluation metrics depends on the

subjective choices and intended use, and is fundamentally a cost-loss problem where the user should arbitrate between the cost of missing exceedances and predicting false alarms.

The performance of the RAW forecasts was found to be only slightly sensitive to the lead day, but this sensitivity was substantially stronger with some MOS methods (although lower than for the persistence method). This aspect is important, although different users may have different needs in terms of lead time, depending on the intended use of the AQ forecast. Forecasts at D+1 may already be useful for some applications like warning in advance the vulnerable population so that they could adapt their outdoor activities. However, implementing short-term emission reduction measures at local scale usually goes through decisions taken at different administrative and political levels, and thus typically requires forecasts at least at D+2. If such measures would have to be taken at larger scale, the occurrence of $O_3$ episodes would probably need to be forecasted even more in advance.

We saw that some forecast methods like PERS or MA can provide a reasonable performance at D+1 but quickly deteriorate when looking further in the future (while other methods like GBM, AN or QM were less impacted by the lead time). Actually, the performance of our PERS(1) reference forecast obviously depends on the typical duration of $O_3$ episodes over the region of study; one (single) episode is defined here as a suite of successive days showing an exceedance of a given threshold at a given station. Over the Iberian Peninsula domain in 2018-2019, considering the target threshold (d8max > 60 ppbv), a total of 6,540 such $O_3$ episodes were observed on the $O_3$ monitoring network with min/mean/max duration of 1/2/27 d (and 5th/25th/50th/75th/95th percentiles of 1.0/1.0/1.0/2.0/5.0 d). Note the 27-d-long $O_3$ exceedance occurred in June-July 2019 at about 30 km north of Madrid (station code *ES1802A*). Considering the information threshold, 240 episodes were observed, with min/mean/max duration of 1/1.1/5 d (and 5th/25th/50th/75th/95th percentiles of 1.0/1.0/1.0/1.0/2.0 d). This may partly explain why the deterioration of performance with lead time was stronger for target thresholds compared to information thresholds.

For operational purposes, several important aspects are to be taken into account. A first aspect concerns the input data required by the MOS method. Does the MOS method rely on observations, models or a combination of both? When the method relies on observations, are they needed in near real-time? How much historical data are required? When the method relies on historical data, to which extent the length of the historical dataset impacts the performance? Related to this last point, another essential aspect concerns the ability of the MOS method to handle progressive and/or abrupt changes in the AQ forecasting system (e.g. configuration, parameterizations, input data like emissions) and/or in the Earth's atmosphere (long-term trends, anomalous events like the COVID-19-related emission reduction, climate change). In this frame, the year 2020 obviously offers a unique large-scale case study to investigate the behavior of the different MOS methods.

MOS methods relying only on very recent data (namely MA and KF methods) are evidently more adaptable to rapid changes, which is a clear asset under changing atmospheric conditions or modeling system configurations. On the other hand, they naturally discard all the potentially useful information available within the historical dataset. Methods like QM, AN or GBM aim at extracting such information to produce better forecasts, but implicitly rely on the assumption that these historical data are still up-to-date and thus representative of the current conditions, which can be a too strong hypothesis when the historical dataset is long or the emission forcing and/or meteorological conditions are changing rapidly. In this study, we considered a

relatively short 2-year dataset but using a longer training dataset would likely require building specific methodologies to tackle this issue, either by identifying and discarding the potentially outdated data, or by giving them a lower weight in the procedure. In this study, we implemented a relatively simple ML-based MOS method. Although the performance on categorical metrics was found limited despite encouraging results on continuous metrics, there is likely room for improvements in near-future developments. In order to improve the high $O_3$ detection skills, potential interesting aspects to explore include testing other types of ML models, customizing loss function and/or cross-validation scores, designing specific weighting strategies and/or re-sampling approaches or comparing regression and classification ML models for the detection of exceedances. Along the preparation of this study, some of them have been investigated but more efforts are required to draw firm conclusions regarding their potential for better predicting $O_3$ episodes. Finally, we focused here on the CAMS regional ensemble but including the individual CAMS models in the set of ML input features may help achieving better performance if the ML model is somehow able to learn the variability (in time and space, or during specific meteorological conditions) of strengths and weaknesses of each model and build its predictions based on the most appropriate sub-set of individual models. More generally, the performance of the different MOS methods is expected to vary from one raw model to another. Investigating the performance and behavior of these methods on the different individual models might shed an interesting light on the results obtained here with the ensemble, and eventually allow generalizing some of our conclusions.

*Data availability.* The EEA AQ e-Reporting and ERA5 dataset used in this study are publicly available.

## Appendix A: Quality assurance with GHOST

Using the metadata available in GHOST (Globally Harmonised Observational Surface Treatment), a quality assurance screening is applied to $O_3$ hourly observations, in which the following data are removed : missing measurements (GHOST's flag 0), infinite values (flag 1), negative measurements (flag 2), zero measurements (flag 4), measurements associated with data quality flags given by the data provider which have been decreed by the GHOST project architects to suggest the measurements are associated with substantial uncertainty or bias (flag 6), measurements for which no valid data remains to average in temporal window after screening by key QA flags (flag 8), measurements showing persistently recurring values (rolling 7 out of 9 data points; flag 10), concentrations greater than a scientifically feasible limit (above 5000 ppbv) (flag 12), measurements detected as distributional outliers using adjusted boxplot analysis (flag 13), measurements manually flagged as too extreme (flag 14), data with too coarse reported measurement resolution (above 1.0 ppbv) (flag 17), data with too coarse empirically derived measurement resolution (above 1.0 ppbv) (flag 18), measurements below the reported lower limit of detection (flag 22), measurements above the reported upper limit of detection (flag 25), measurements with inappropriate primary sampling for preparing $NO_2$ for subsequent measurement (flag 40), measurements with inappropriate sample preparation for preparing $NO_2$ for subsequent measurement (flag 41) and measurements with erroneous measurement methodology (flag 42).

## Appendix B: Kalman filter

In this section, we briefly describe the application of the Kalman filter as a MOS correction method. More details can be found for instance in Delle Monache et al. (2006), while Pei et al. (2017) provides a clear general introduction to the Kalman filter. CAMS forecasts are available over 4 lead days, from D+1 to D+4. We define here the time $t$ as the day D at a given hour of the day ($t+1$ thus corresponds to D+1 at this specific hour of the day). In an operational context, observations at this hour of the day are available only until time $t$ (included). In this frame, our primary objective in this MOS approach is to estimate $x_{t+1|t}$, the true (unknown) forecast bias at time $t+1$ using the information available until $t$ (included), which can then be used to correct the raw CAMS forecast. Here, $x_{t+1|t}$ can be referred to as the a priori forecast bias at time $t+1$ while $x_{t+1|t+1}$ can be referred to as the a posteriori forecast bias at time $t+1$ as it takes benefit from the information obtained at $t+1$. We distinguish estimated values from true values using an hat (^) ($\hat{x}_{t+1|t}$ therefore corresponds to the estimated value of $x_{t+1|t}$). In its application as a MOS method, the Kalman filter considers the following *process equations* for describing the time evolution of the forecast bias:

$$x_{t+1|t} = x_{t|t} + \eta_{t+1} \; ; \; (\hat{x}_{t+1|t} = \hat{x}_{t|t}) \tag{B1}$$

$$p_{t+1|t} = p_{t|t} + \sigma_\eta^2 \tag{B2}$$

where $\eta_{t+1}$ represents the process noise and is assumed to be a white noise term with normal distribution, zero-mean, variance $\sigma_\eta^2$ and uncorrelated in time, and $p_{t+1|t}$ the a priori expected error variance of the forecast bias estimate. Our process equations here are thus quite simple as we assume that the a priori forecast bias at time $t+1$, $x_{t+1|t}$, is similar to the previous a posteriori forecast bias $x_{t|t}$ but with some uncertainty $\eta_{t+1}$.

At time $t+1$, an observation of the forecast bias $x_{t+1}$, denoted $z_{t+1}$, is available but with some uncertainty (since the measurement of the pollutant concentration necessarily comes with some uncertainty):

$$z_{t+1} = x_{t+1} + \epsilon_{t+1} \tag{B3}$$

where $\epsilon_{t+1}$ represents the measurement noise and is assumed to be a white noise term with normal distribution, zero-mean, variance $\sigma_\epsilon^2$ and uncorrelated in time, and independent from the process noise $\eta_{t+1}$. Then, the Kalman filter allows to fuse this observation $z_{t+1}$ and the a priori estimate of the forecast bias $x_{t+1|t}$, in order to obtain an a posteriori estimate of the forecast bias $x_{t+1|t+1}$ :

$$K_{t+1} = (p_{t+1|t} + \sigma_\eta^2)/(p_{t+1|t} + \sigma_\eta^2 + \sigma_\epsilon^2) \tag{B4}$$

$$\hat{x}_{t+1|t+1} = \hat{x}_{t+1|t} + K_{t+1}(z_{t+1} - \hat{x}_{t+1|t}) \tag{B5}$$

$$p_{t+1|t+1} = (p_{t+1|t} + \sigma_\eta^2)(1 - K_{t+1}) \tag{B6}$$

where $K_{t+1}$ corresponds to the so-called Kalman gain used to weight the respective importance of the a priori forecast bias estimate ($\hat{x}_{t+1|t}$) and its observed value ($z_{t+1}$), and $p_{t+1|t}$ the expected error of the forecast bias estimate (i.e. the variance of the forecast bias error : $p_{t+1|t} = Var(x_{t+1|t} - \hat{x}_{t+1|t})$).

In practise, the KF algorithm first requires initializing the $\hat{x}_{0|0}$ and $p_{0|0}$ values (any reasonable value can be chosen, given that the KF quickly converges). Then the algorithm starts its first iteration. As a first step, the a priori estimated value of the forecast bias $\hat{x}_{1|0}$ is obtained from $\hat{x}_{0|0}$ (in our problem, we simply have : $\hat{x}_{t+1|t} = \hat{x}_{t|t}$) and used to correct the raw forecast of CAMS. As a second step, after obtaining the observed pollutant concentration, one can deduce $z_1$ and fuse it with $\hat{x}_{1|0}$ using the Kalman filter equations, which gives us the a posteriori estimated value of the forecast bias $\hat{x}_{1|1}$, that will be available for the second iteration. An overview of this workflow is given in Fig. B1.

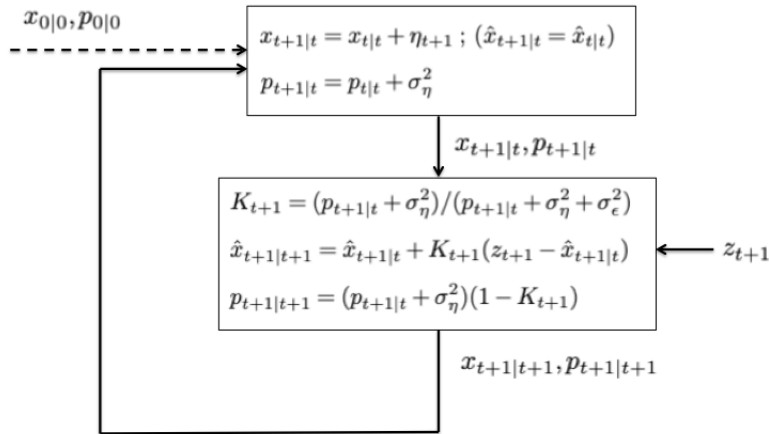

**Figure B1.** Workflow of the Kalman filter method.

Solving these equations requires assigning values to both variances $\sigma_\eta^2$ and $\sigma_\epsilon^2$. It can be demonstrated that, once $\sigma_\epsilon^2$ is set to a fixed value (any reasonable value can be chosen, for instance $\sigma_\epsilon^2 = 1$), the KF results mainly depend on the $\sigma_\eta^2/\sigma_\epsilon^2$ variance ratio. Various strategies can be used to choose an appropriate value for this variance ratio. This aspect is discussed in Sect. 2.3.3.

## Appendix C: Analogs norm

The analogs (AN) method requires to identify which past forecast days are the most similar to the current one. Given a set of features to take into account, this similarity is computed using the norm introduced by (Delle Monache et al., 2006) :

$$\|F_t, A_{t'}\| = \sum_{i=1}^{N} \frac{w_i}{\sigma_i} \sqrt{\sum_{t=-T}^{T} (F_{i,t+k} - A_{i,t'+k})^2} \tag{C1}$$

with $F_t$ the raw forecast at time $t$, $A_{t'}$ an analog forecast at time $t'$, $N$ the number of features taken into account, $w_i$ the weight of the feature $i$, $\sigma_i$ its standard deviation calculated over past forecasts, $T$ the half the width of the time window over which to

705 compute the metric (i.e. a value $T = 2$ means that the squared difference between the forecast and the analog will be computed over a $\pm2$ hours time window). In our study, we used weights of 1 for all features (wind speed, wind direction, temperature, surface pressure) and $T = 1$.

## Appendix D: Tuning of the GBM models

The GBM models are tuned using a so-called *randomized search* in which a range of values is given for each hyperparameter
of interest and a total number of hyperparameters combinations to test. After fixing the learning rate to 0.05 (*learning_rate* in the *scikit-learn* Python package), the tuning of the GBM model was done over the following set of hyperparameters: the tree maximum depth (*max_depth* : from 1 to 5 by 1), the subsample (*subsample* : from 0.3 to 1.0 by 0.1), the number of trees (*n_estimators*: from 50 to 1000 by 50) and the minimum number of samples required to be at a leaf node (*min_samples_leaf*: from 1 to 50). As we are dealing here with time series, this tuning is conducted through a rolling-origin cross-validation in
which validation data are always posterior to train data.

## Appendix E: Evaluation metrics

The continuous metrics used in this study are defined as followed :

$$\mathrm{MB} = \frac{1}{N} \sum_{i=1}^{N} m_i - o_i \tag{E1a}$$

$$\mathrm{nMB} = \frac{\mathrm{MB}}{\overline{o}} \tag{E1b}$$

$$\mathrm{RMSE} = \sqrt{\frac{\sum_{i=1}^{N}(m_i - o_i)^2}{N}} \tag{E1c}$$

$$\mathrm{nRMSE} = \frac{\mathrm{RMSE}}{\overline{o}} \tag{E1d}$$

$$\mathrm{PCC} = \frac{1}{N-1} \sum_{i=1}^{N} \frac{(m_i - \overline{m})(o_i - \overline{o})}{\sigma_m \sigma_o} \tag{E1e}$$

$$\mathrm{nMSDB} = \frac{1}{N} \sum_{i=1}^{N} \sigma_i - \sigma_i \tag{E1f}$$

with $m_i$ and $o_i$ the predicted and observed mixing ratios, $\overline{m}$ and $\overline{m}$ their corresponding mean, $\sigma_m$ and $\sigma_m$ their corresponding
standard deviation, and $N$ the number of points.

The performance of the categorical forecasts of exceedances beyond a certain threshold can primarily be described through a contingency table (Tab. E1). Based on these individual numbers $a$ (hits), $b$ (false alarms), $c$ (misses) and $d$ (correct rejections), a wide number of verification metrics have been proposed in the literature, often with inconsistent nomenclature. In order to avoid confusions, all metrics used in this paper systematically follow the nomenclature given in the reference book of Jolliffe
and Stephenson (2011).

**Table E1.** Schematic contingency table for deterministic forecasts of binary exceedances of the regulatory limit values.

| Exceedance forecast | Exceedance observed | | |
|---|---|---|---|
| | Yes | No | Total |
| Yes | $a$ (hits) | $b$ (false alarms) | $a+b$ |
| No | $c$ (misses) | $d$ (correct rejections) | $c+d$ |
| Total | $a+c$ | $b+d$ | $a+b+c+d=n$ |

For a given total number of data $n$ $(=a+b+c+d)$, the 2x2 contingency table can be fully described by three independent measures, namely the base rate $s$ independent from the forecasting system (total proportion of observed exceedances, also known as the climatological probability of an exceedance), the hit rate $H$ (proportion of the observed exceedances that are correctly detected) and the false alarm rate $F$ (proportion of the observed non-exceedances erroneously forecast as exceedances, to be distinguished from the false alarm ratio). These metrics as well as the other categorical metrics used in this study - Frequency Bias (FB), Success Ratio (SR), Critical Success Index (CSI) or Peirce Skill Score (PSS) - are defined as follows:

$$s = (a+c)/n \tag{E2a}$$

$$H = a/(a+c) \tag{E2b}$$

$$F = b/(b+d) \tag{E2c}$$

$$PC = (a+d)/n = (1-s)(1-F) + sH \tag{E2d}$$

$$FB = (a+b)/(a+c) = (1-s)F/s + H \tag{E2e}$$

$$SR = (a)/(a+b) = 1 - \left[1 + \left(\frac{s}{1-s}\right)\frac{H}{F}\right]^{-1} \tag{E2f}$$

$$CSI = a/(a+b+c) = \frac{H}{1+F(1-s)s} \tag{E2g}$$

$$PSS = \frac{ad-bc}{(b+d)(a+c)} = H - F \tag{E2h}$$

Note that as shown in these formula, any categorical metric initially function of $a$, $b$, $c$ and $d$ can be expressed in terms of $s$, $H$ and $F$. One interest of considering this $s$-$H$-$F$ framework (so-called likelihood-base rate factorization, see chapter 3 of Jolliffe and Stephenson (2011) for a detailed description) lies in the fact that, since the forecaster does not have any influence on $s$, the tri-dimensional problem is reduced to bi-dimensional ($H$ and $F$). Since it is easily possible to maximize $H$ (by always predicting an exceedance) or $F$ (by always predicting a non-exceedance), none of these two metrics taken individually is a good and balanced metric for assessing the quality of a forecasting system; only some combinations of both (eventually with $s$) can eventually provide a good way to assess this detection skills, such as those used in this study.

## Appendix F: Time series

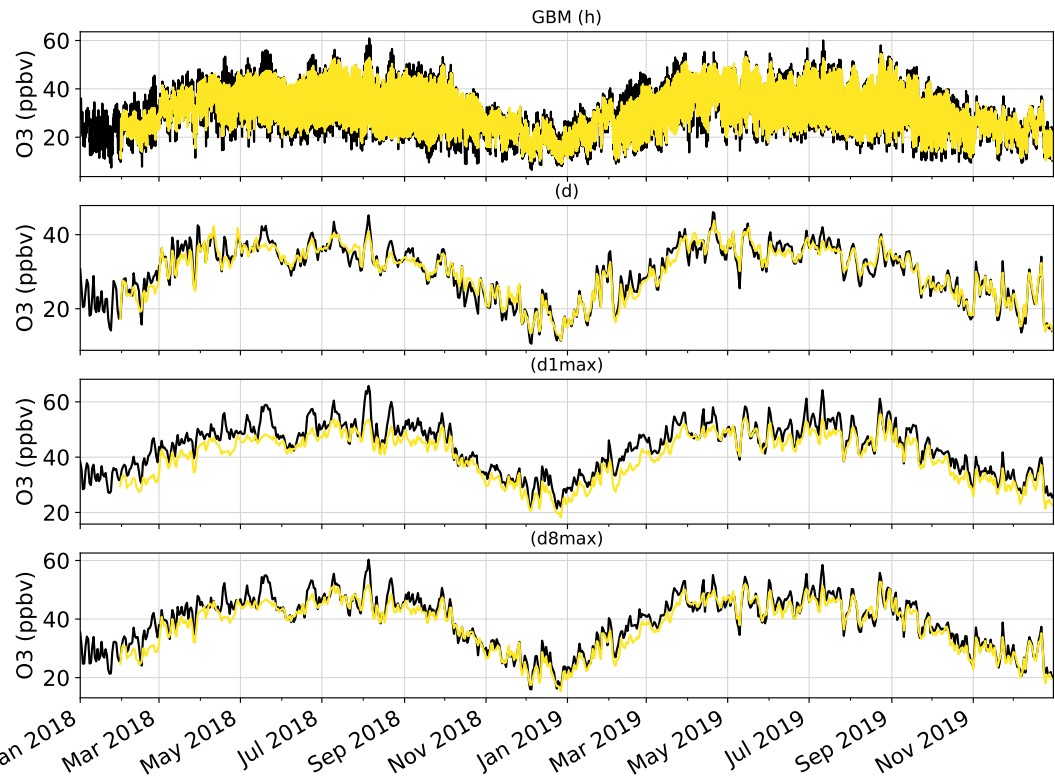

**Figure F1.** Time series of the mean O$_3$ mixing ratios over the Iberian Peninsula, as observed by monitoring stations (in black) and as simulated by CAMS D+1 forecasts corrected with the GBM MOS method (in yellow). Time series are shown at the hourly (h), daily mean (d), daily 1-hour maximum (d1max) and daily 8-hour maximum (d8max) time scales. O$_3$ mixing ratios are averaged over all surface stations of the domain.

## Appendix G: Sensitivity tests

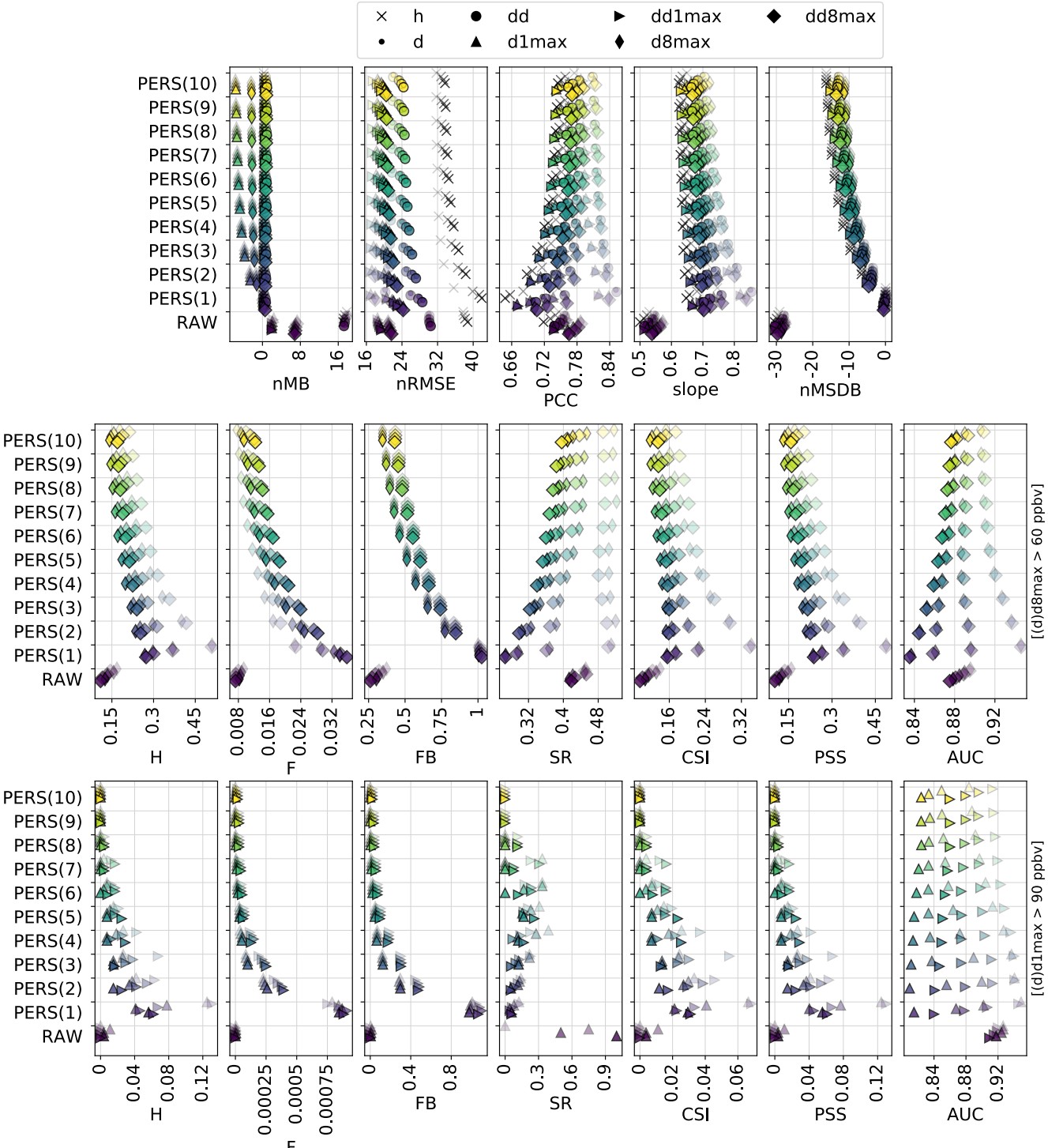

**Figure G1.** Similar to Fig. 3 for sensitivity tests on the PERS method.

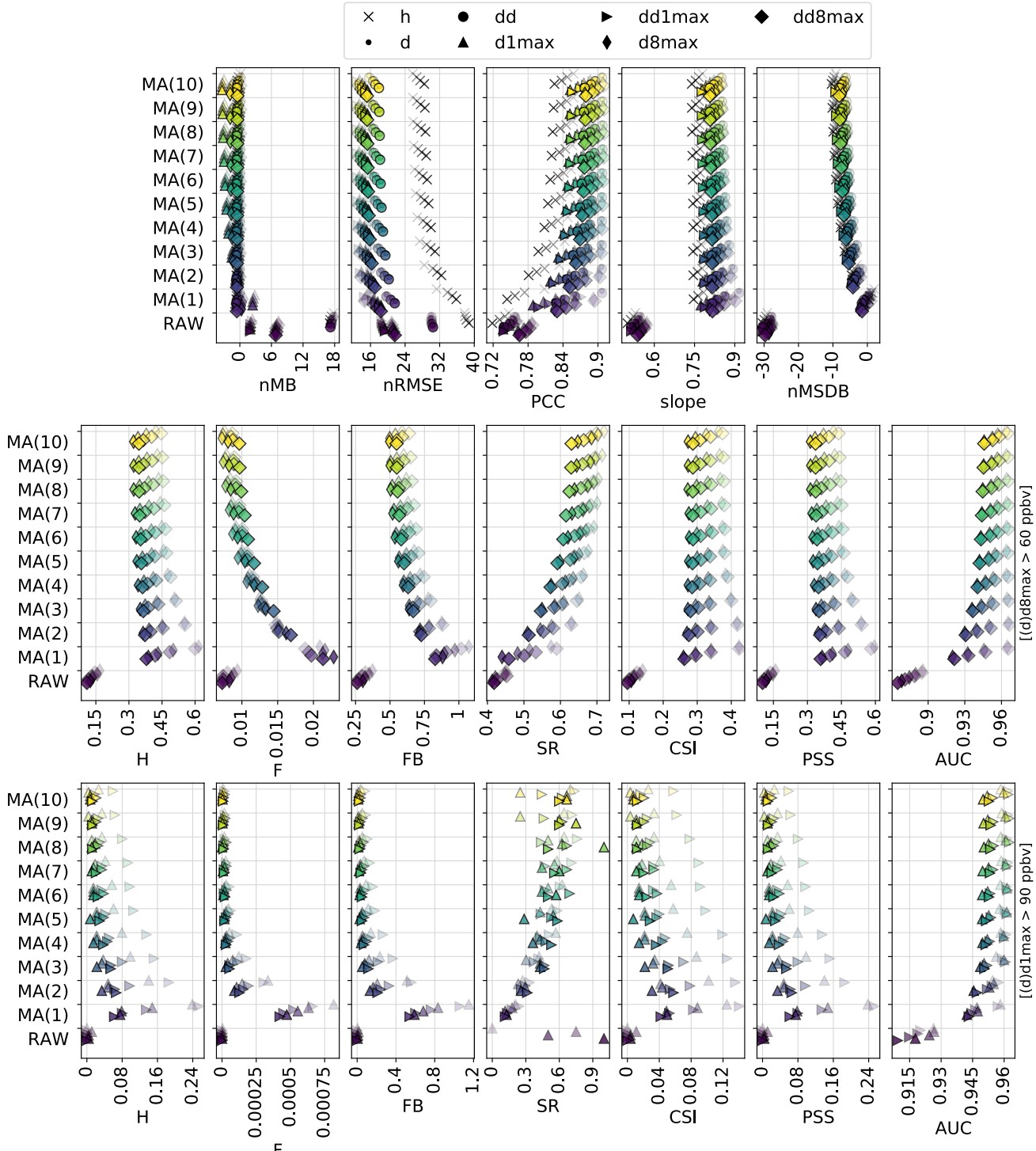

**Figure G2.** Similar to Fig. 3 for sensitivity tests on the MA method.

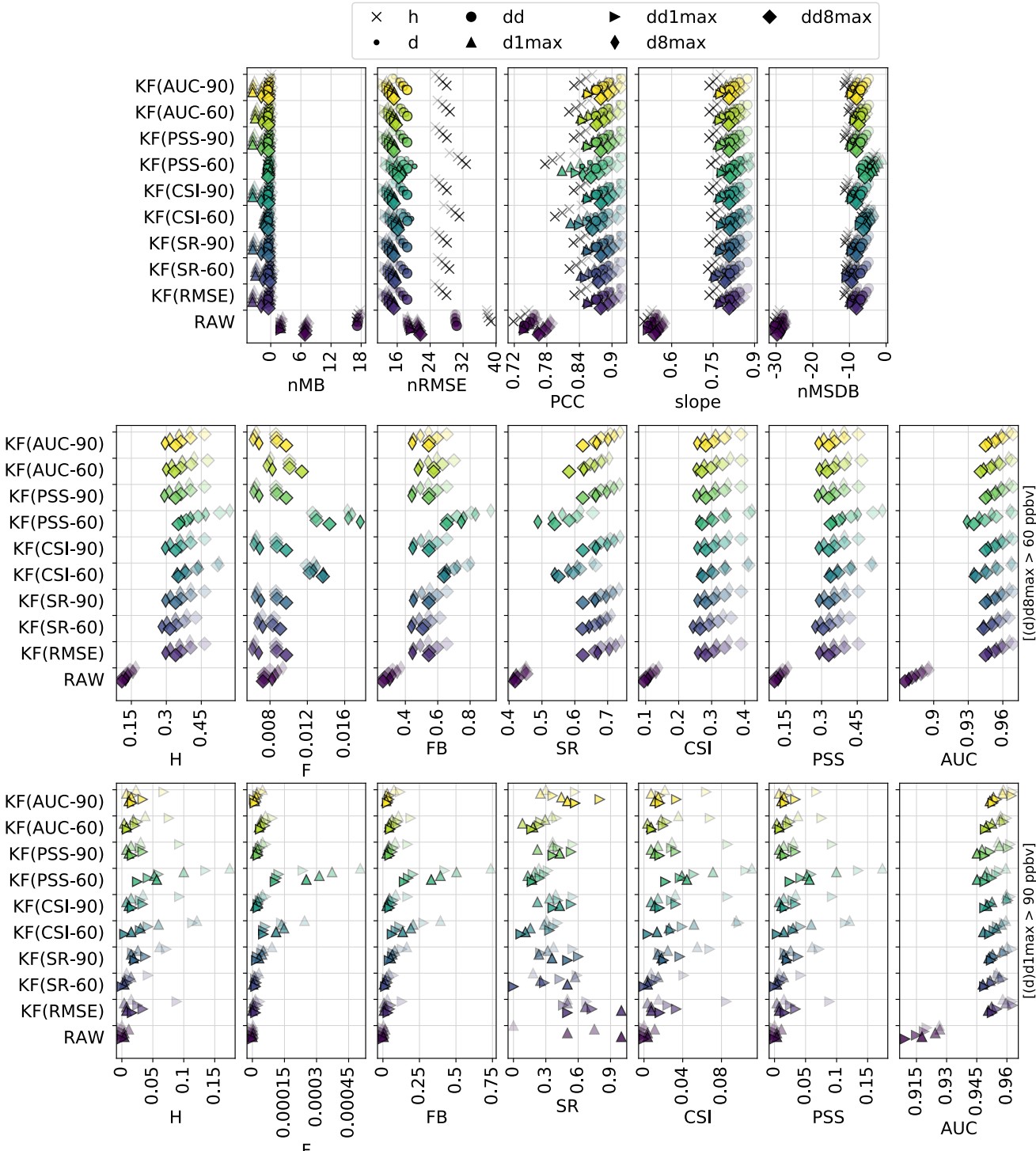

**Figure G3.** Similar to Fig. 3 for sensitivity tests on the KF method.

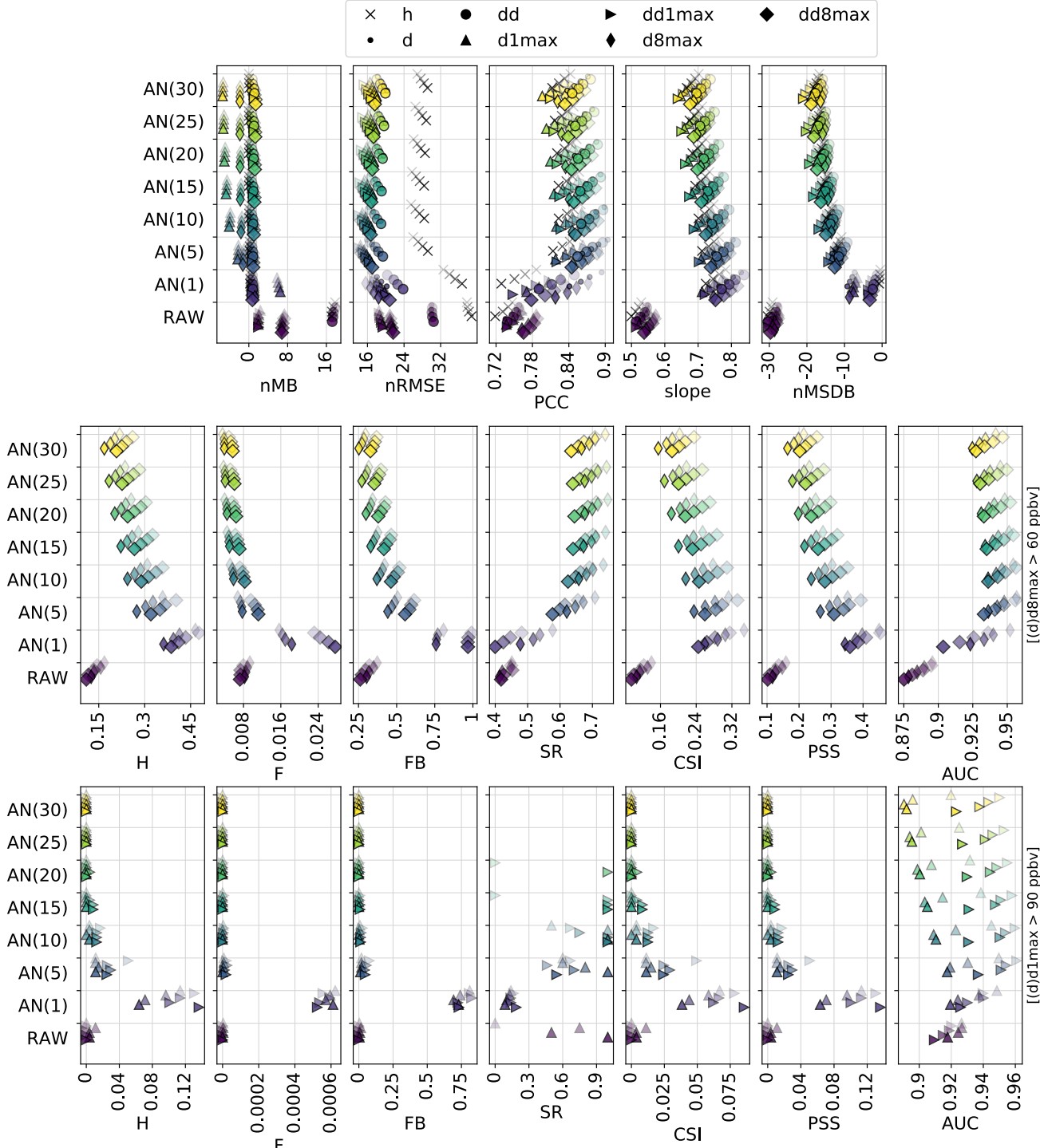

**Figure G4.** Similar to Fig. 3 for sensitivity tests on the AN method.

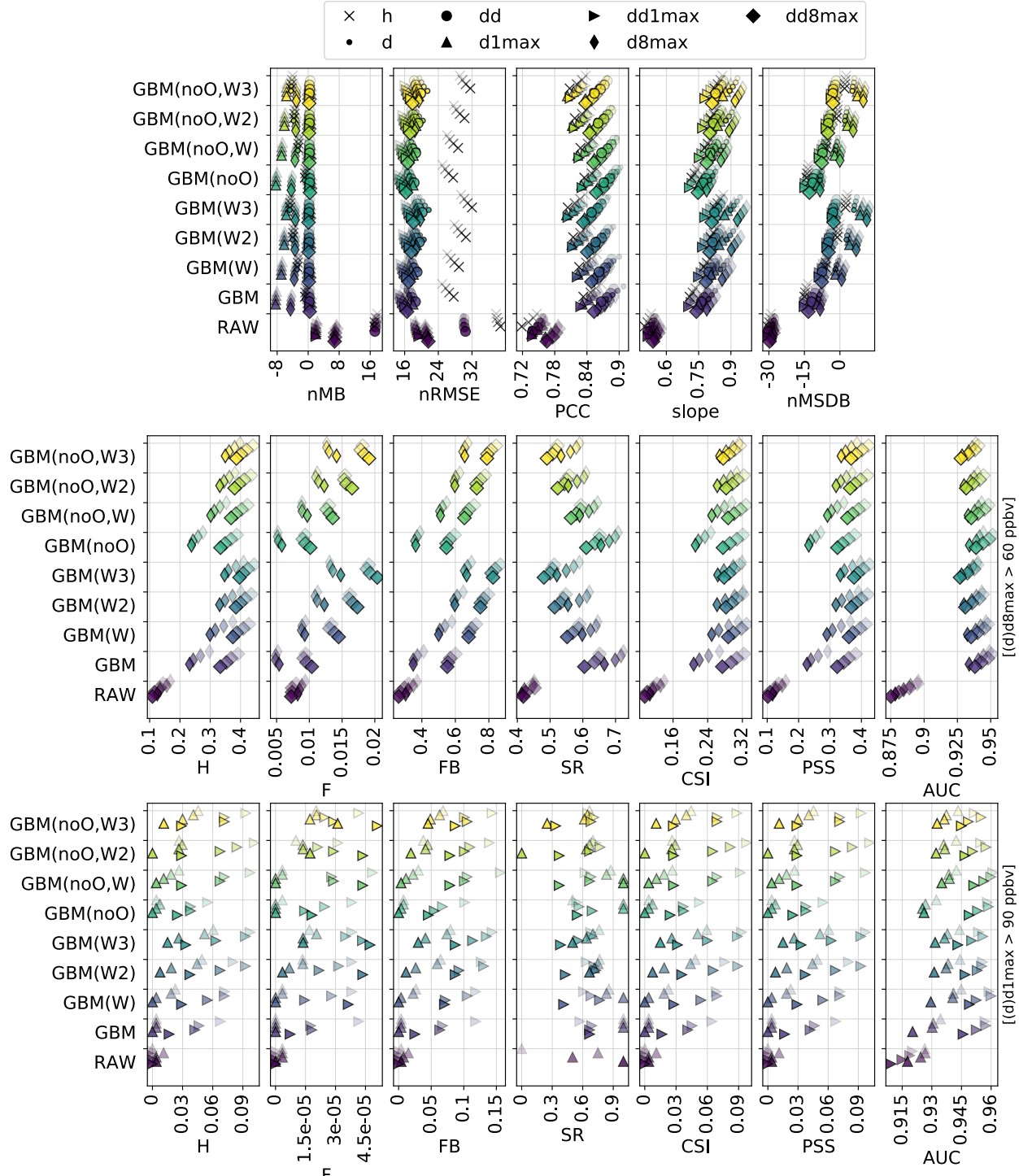

**Figure G5.** Similar to Fig. 3 for sensitivity tests on the GBM method.

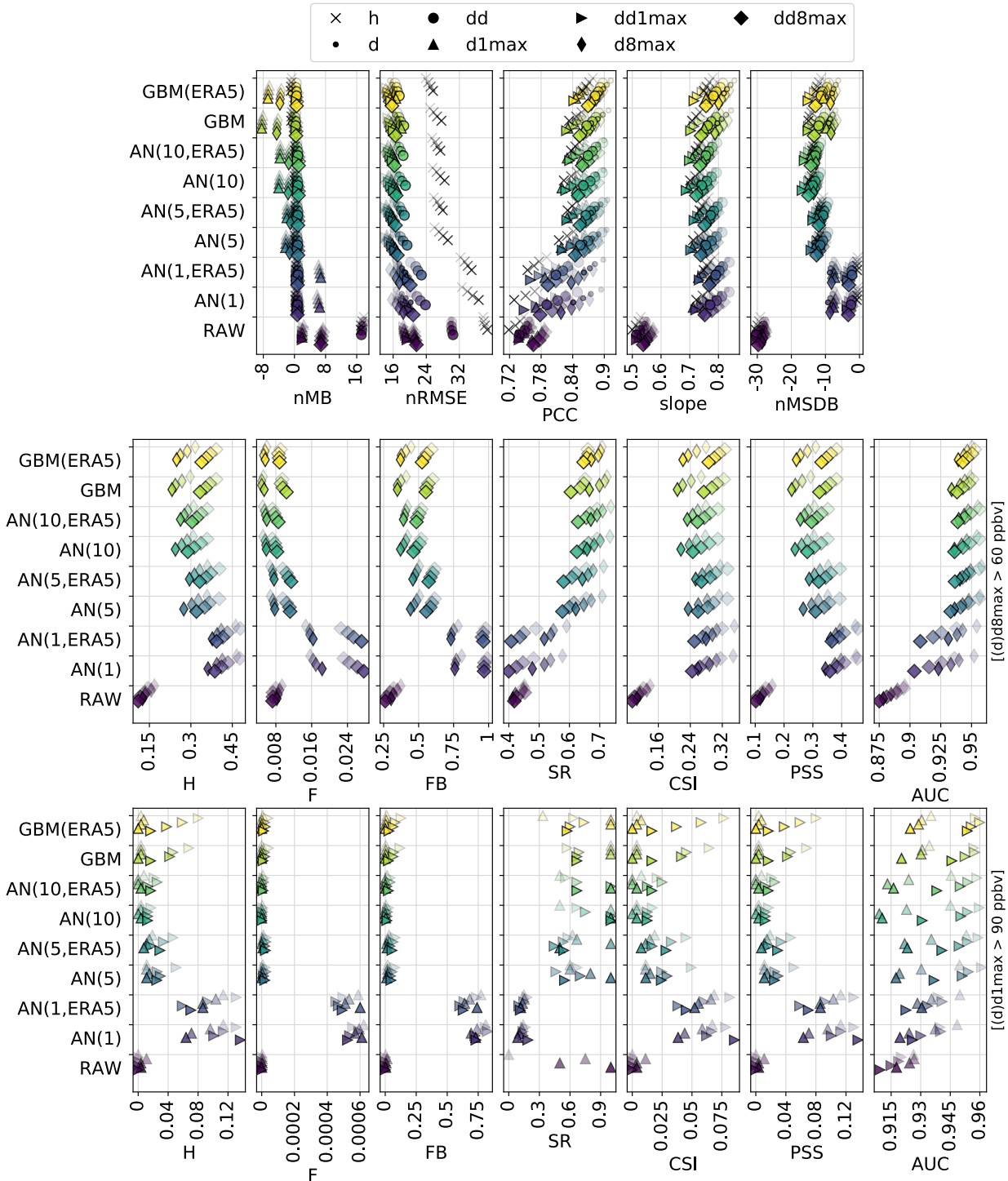

**Figure G6.** Similar to Fig. 3 for sensitivity tests on the meteorological data (HRES versus ERA5) used in AN and GBM methods.

*Author contributions.* HP contributed to the conception and design of the study. PAB and MSC were responsible for downloading the CAMS and meteorological data. KS was responsible for installing the python packages and other useful modules on the Mare Nostrum supercomputer. DB was responsible for the acquisition and preprocessing of the air quality data through the GHOST project. HP carried out the analysis. HP, CPGP, OJ, AS, MG, JMA and DB contributed to the interpretation of results. HP was responsible for writing the article, with a careful review from CPGP and JAM.

*Competing interests.* The authors declare that they have no conflict of interest.

*Acknowledgements.* This research has been funded by the European Union's Horizon 2020 research and innovation programme under the Marie Sklodowska-Curie grant agreement H2020-MSCA-COFUND-2016-754433, as well as the MITIGATE project (PID2020-116324RA-I00 / AEI / 10.13039/501100011033) from the Agencia Estatal de Investigacion (AEI). We also acknowledge support by the the AXA Research Fund and Red Temática ACTRIS España (CGL2017-90884-REDT), the BSC-CNS "Centro de Excelencia Severo Ochoa 2015-2019" Program (SEV-2015-0493), PRACE and RES for awarding us access to Marenostrum Supercomputer in the Barcelona Supercomputing Center, and H2020 ACTRIS IMP (#871115).

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
