# Peer review of "Model Output Statistics (MOS) applied to CAMS O3 forecasts: trade-offs between continuous and categorical skill scores"

_Atmospheric Chemistry and Physics, 2021_

## Author Comment (AC1)

We are thankful for the numerous constructive comments and suggestions provided by the two reviewers. Below, the comments of the reviewers are indicated in grey, our answers in black and the modifications of the text in blue.

**Reviewer #1**

**Overview**

The paper compares different MOS methods applied to the regional CAMS forecast at stations locations over the Iberian Peninsula for ozone in 2018-19. It uses an "operational scenario approach", namely that observations become only gradually available to learn the different methods. The paper finds that the MOS approaches have different strength and weaknesses with respect to improvements of the overall reduction of the forecast error and the error specifically for high pollution episodes and threshold exceedances.

**General comments**

The paper reports about a sound scientific effort, in particular the consideration of accuracy measures in general and for exceedances of threshold during episodes of high pollutions is very welcome. But, some methodological aspects need to be reconsidered. Also, the result, method and choice of accuracy measures are not explained with enough detail, which would be required to better understand the results and applicability of the different approaches. The paper does not present well the large amount of information coming from the combination of the many MOS approaches and accuracy measures. The authors need to introduce tables showing the accuracy measures for each MOS type, which allows the reader to digest the information. Tables could also help to substantially shorten the long narrative descriptions.

As discussed in more details below, we proposed in the revised version numerous modifications that will hopefully improve the manuscript along the lines suggested by reviewer #1. We added a new section to present the different metrics used (while at the same time reducing the corresponding section in the Appendix) and to introduce the corresponding skill scores to be discussed.

Regarding tables, in the initial version of the manuscript we decided on purpose to avoid including tables essentially because the results shown in Fig. 3 are multidimensional (for each statistical metric : MOS method, lead time, time scale). Thus, to our opinion, tables are not necessarily the most convenient way to present such a large quantity of results as they would not allow the different metrics to be easily compared among each other (as this would require many tables). However, we agree that some tables can facilitate the reading and shorten the discussion. Therefore, we selected a subset of these statistics and included them into two tables. We included additional tables in the Supplement (notably for the different sensitivity tests).

Also, as shown below, we shortened the narrative descriptions while analyzing in more detail the results shown in Fig. 3.

On the other hand, sensitivities to input parameters and variation of the methods are discussed with some detail, which make the paper somewhat unbalanced. Although interesting in itself, it is also not clear what the purpose of that section is. Are the results presented in 3.3 and 3.2 already carried out with the optimal choice of parameter setting or not ? The discussion in 3.4. should be shortened by focusing on application with a very high sensitivity to parameter choice.

In the revised version of the manuscript, we reorganized the discussion of the results in Sect. 3.2 and 3.3 (see below) and shortened the discussion on the sensitivity tests (section 3.4), which should improve the balance of the paper. Regarding this last section, although it might disrupt a bit the general narrative of the paper, we do think it is important to show the sensitivity of these different methods to their internal parameters. The methods used in Sect. 3.2 and 3.3 are not considering an optimal choice of internal parameters for the simple reason already raised in the discussion that there is no optimal MOS methods without a proper definition of user-specific needs and interests (in other words, a clear choice of the metric of strongest interest). We understand the frustration that the absence of clear recommendations might create but this is one of the conclusions of our study (as reflected in the title we choose): the behavior of the different MOS methods strongly varies with the choice of metric, for instance with some MOS methods showing the best continuous metrics and the worst categorical ones. Therefore, in an operational context, the choice of the MOS method (and its internal parameters) directly depends on the desired behavior of the forecast system (Is the user more interested in having forecasts with lowest bias and error or in predicting exceedances? For this latter category, is the user more interested in avoiding false alarms at the potential cost of missed episodes, or is he interested in a more precautionary approach where more false alarms are accepted?).

We applied the following modifications in the introduction of section 3.4 :

L333 : "In the previous sections, we provided a first evaluation of the performance of a set of MOS methods. All methods rely on specific choices or parameters that can substantially influence the behavior of the MOScorrected forecasts, and thus its general performance. In this section, we discuss some of these choices and investigate their impact on the performance through different sensitivity tests."  $\rightarrow$  "Each of the forecast methods considered in this study relies on a specific configuration, e.g. the time window of PERS or MA methods, the metric used internally in KF for optimizing the variance ratio, the number of analogs taken into account in AN, the choice of input features or metrics used internally for fitting the ML model in GBM. This configuration can substantially influence their general performance, although in a different way depending on the metric used. In the previous sections, we evaluated the performance of these different methods considering a relatively simple baseline configuration. In this section, we discuss some of these choices and investigate their impact on the performance through different sensitivity tests. Corresponding statistical results on continuous and categorical metrics are given in Tables in the Supplement."

**And we slightly shortened some of subsections :**

• L338 : "The persistence method essentially relies on the choice of the time window over which past observations are averaged to provide the O3 forecast. In the previous section, we used a window of 1 d. A sensitivity test is performed with windows ranging between 1 and 10 d (hereafter referred to as PERS(n) with n the window in days). Results are shown in Fig. G1 in Appendix G, and indicate that, while PERS(1) forecasts were unbiased (whatever the time scale), increasing the window leads to a growing negative bias on d1max and d8max scales. The bias is substantially reduced when working at dd1max and dd8max scales, i.e. when applying the PERS approach directly on daily 1-hour and 8-hour maximums rather than on the hourly time series. The differences between the two approaches originate from the dayto-day variability in the hour of the day when O3 mixing ratios peak. For illustration purposes, let's assume that O3 peaks between 15 and 17 h; on a given day, O3 mixing ratios at 15/16/17h reach 50/60/50 ppbv and on the following day 70/70/80 ppbv. Then, the PERS(2)dd1max O3 would be 70 ppbv (mean of 60 and 80 ppbv), while the PERS(2)d1max O3 would be only 65 ppbv (maximum of the mean diurnal profile of these two days, in this case 60/65/65).

[revised manuscript text omitted]

- L430 : "In this context, it appears interesting to evaluate to which extent the performance is altered when not relying on this specific information. Results are shown in Fig. G5 in the Appendix G."
- L436 : "Regarding the skills for detecting d8max O3 above 60 ppbv, stronger weights typically increase both H and F, improve the (underestimated) FB, but deteriorate the SR and AUC (the forecasts become more liberal). Regarding the more balanced metrics (of strongest interest here), adding more weights on the tails of the O3 distribution has a positive although small impact on PSS. A minor positive impact is also found on CSI, but the best

results are obtained with GBM(W2), thus moderate weights. For both metrics, improvements are most obvious at the d8max scale, while changes at the dd8max scale are much smaller. Regarding the detection of d1max O3 above 90 ppbv, the influence of the weighting strategies is more ambiguous but the detection skills generally remain very poor. Again, the strongest CSI or PSS improvements are obtained at the d1max scale with much lower changes of the dd1max results." → "Regarding the skills for detecting target threshold exceedances, stronger weights typically increase both H and F, improve the (underestimated) FB, but deteriorate the SR and AUC (the forecasts become more liberal). Regarding the more balanced metrics (of strongest interest here), adding more weights on the tails of the O3 distribution typically has a positive although small impact on CSI and PSS. Regarding the detection of information threshold exceedances, both CSI and PSS can also be slightly improved by adding some weight into the GBM, but the performance for detecting such high O3 values remain relatively low. The interest of using the O3 concentration observed one day before is here found to be limited."

It remains unsatisfactory to treat the persistence approach as a variant of MOS. As the author explain themselves, persistency is a reference forecast (to identify if a given forecast has skill compared to the reference) and both the RAW as well as the other MOS approaches should be more directly compared against it. An important question for all forecast application is, if RAW beats PERS (depending on the accuracy measure) and if and how MOS (using RAW) can improve the skill.

We agree with the reviewer regarding the specificity of the PERS method, that cannot be considered as an additional MOS method. In the revised version of the manuscript, we modified the text to avoid confusion regarding this aspect. Concerning the presentation of the results, we consider it is useful to keep showing and discussing the different metrics for the RAW, the PERS and the different MOS-corrected forecast taken individually as done in the initial version. However, in the revised version, we added some extra discussion of the results in terms of skill scores, taking the PERS(1) forecast as a reference. We included a new figure equivalent to Fig. 3 showing the corresponding skills scores.

**We applied the following modifications :**

- L4 : "In this study, we investigate to what extent AQ forecasts can be improved using a variety of MOS methods, including persistence (PERS), moving average (MA), quantile mapping (QM), Kalman Filter (KF), analogs (AN), and gradient boosting machine (GBM)." → "In this study, we investigate to what extent AQ forecasts can be improved using a variety of MOS methods, including moving average), quantile mapping, Kalman Filter, analogs, and gradient boosting machine, and consider as well the persistence method as a reference."
- L126 : "This section describes the different MOS methods implemented for correcting the raw forecasts (hereafter referred to as RAW), namely: persistence (PERS), moving average (MA), Kalman filter (KF), quantile mapping (QM), analogs (AN) and gradient boosting machine (GBM). All MOS

methods are applied independently on each monitoring station."  $\rightarrow$  "This section describes the different MOS methods implemented for correcting the raw forecasts (hereafter referred to as RAW), namely: moving average (MA), Kalman filter (KF), quantile mapping (QM), analogs (AN) and gradient boosting machine (GBM). All MOS methods are applied independently on each monitoring station. The skill of these different forecasts (including the RAW) is assessed relative to the Persistence (PERS) reference method, which uses the previously observed concentration values at a specific hour of the day (averaged over 1 or several days) as the predicted value. As a first approach, we use a time window of one single day (hereafter referred to as PERS(1))."

- L129 : "2.3.1 Persistence (PERS) and moving average (MA) methods" → "2.3.1 Moving average (MA) methods"
- L130 : "We primarily consider two relatively simple MOS methods: the persistence (PERS) and the moving average (MA). The PERS method simply uses the previous observed concentrations values at a specific hour of the day (averaged over 1 or several days) as the predicted value for this specific hour. It is often used as a reference to measure the skill achieved by other methods, especially for very short-term forecasts. In the MA method, the forecast bias in the previous day or days is used to correct the forecast. As a first approach, we use a time window of one single day for both PERS and MA methods. The corresponding approaches are hereafter referred to as PERS(1) and MA(1). The sensitivity of both PERS and MA methods to the time window is discussed in Sect. 3.4.." → "We primarily consider the Moving Average (MA) method, by which the raw CAMS forecast bias in the previous day(s) is used to correct the forecast. As a first approach, we use a time vindow of one single day (bereafter referred to as MA(1)). The sensitivity of both PERS and MA methods to the time window of one single day (hereafter referred to as MA(1)). The sensitivity to the time window of one single day (hereafter referred to as MA(1)). The sensitivity to the time window is discussed in Sect. 3.4.".

**We added a dedicated section to describe the evaluation metrics and the corresponding skill scores (and removed the corresponding text in L226-236) : L221 : "2.5 Evaluation metrics and skill scores**

In this study, O3 forecasts are evaluated using an extended panel of continuous and categorical metrics to provide a comprehensive view of the impact of the different MOS methods on the predictions. Continuous metrics used to evaluate the O3 concentrations include :

- nMB : normalized Mean Bias
- nRMSE : normalized Root Mean Square Error
- PCC : Pearson correlation coefficient
- Slope : slope of the predicted-versus-observed O3 mixing ratio, to quantify how well lowest and highest O3 concentrations are predicted
- nMSDB : normalized Mean Standard Deviation Bias, to investigate how well the O3 variability is reproduced by the forecast

Categorical metrics used to evaluate the O3 exceedances beyond certain thresholds include :

[revised manuscript text omitted]

The AQ observations are used without discrimination of the representativeness for the scale of model grid boxes of the regional ensemble (10km). One would expect that some stations (i.e. rural, urban) are more representative than others (i.e. traffic). It is a missed opportunity of the paper to discuss the amount of correction by MOS for the different air quality observations stations based on the station type.

Indeed, such relatively coarse spatial resolution does not allow to properly represent the pollutant concentrations observed at stations close to strong emission sources (e.g. urban traffic, industrial), as already mentioned in the text (L257 : "Part of this positive nMB and negative nMSDB is expected since this broad comparison includes all station types, including traffic stations where local road transport NOx emissions can strongly reduce the O3 levels (titration by NO), which cannot be fully represented by models at 10 km spatial resolution."). In this study, we chose to consider all stations because we are ultimately interested in predicting O3 exceedances at all locations where observations are available and therefore where air quality standards apply. Given the numerous aspects already covered, we consider that it is beyond the scope of this study to explore in more detail the impact of the station types. We added a brief comment :

L260 : "In this study, all station types are included because we are ultimately interested in predicting O3 exceedances at all locations where they can be observed (and thus, where the air quality standards apply). It is worth noting that the impact

**of the MOS methods on the different skills might vary from one type of station to another, although this aspect is beyond the scope of our study."**

The assumption about an operational scenario (observations become gradually available after the start of the application on 1.1.2018) is in principle a welcome approach but several questions remain. It is unclear how different spin-up times (i.e. the time until further improvements by adding more previous data become very small) of the methods, which should also be stated more clearly, are taken into account in the evaluation. Second, it remains unclear what happens in the case, that observations are not available in near-real-time to be fed in to the MOS scheme. Consequently ,it is more important from an operational point of view to apply MOS approaches for the case that observations are always available in NRT or that they are not, which means that these MOS approaches could only be trained with past data. The latter is a typical cross-validation approach, which uses one data set to train and the other to evaluate the MOS. The impact of missing data needs to be discussed in more detail.

Again, the reviewer is raising here interesting questions but the application of MOS methods to operational air quality forecasts is a vast topic of research and we do not claim in this first study to cover all its relevant aspects. To our opinion, regarding the application of MOS methods to air quality forecasts, this work already goes beyond most of the other studies available in the literature, and therefore we consider the points raised by the reviewer as important aspects but beyond the scope of the present study. Nonetheless, according to the reviewer comments we discussed these different points in the revised version :

L124 : "We note, however, that methods relying on limited past data may respond better to an abrupt change in environmental conditions, as experienced for instance during the COVID-19 lockdowns. Although not covered by the present study, we acknowledge here that in an operational context, the relationship between the length of past training data and the performance of the corresponding MOS prediction is an interesting aspect to investigate, as is the quantification of the spinup time beyond which the MOS method might not significantly improve. Only some insights will be given by comparing the performance obtained in 2019 with and without using the data available in 2018. Similarly, our study does not investigates how potential issues (delays) in the near-real time availability of the observations can impact the performance of the MOS methods, although this might be another important aspect to take into account in operational conditions; to the best of our knowledge, EEA observations are typically available with a 2-h lag but some sporadic technical failures can induce extended delays."

It does not make sense to use ER5 as a reference meteorological data set with respect to the HRES NWP forecast in this application. The HRES (IFS) forecast (9km) should be compared against HRES analysis that were the initial conditions of the forecast (step=0) (Both HRES and ER5 are produced with the IFS)

We thank the reviewer for clarifying this point. However, to the best of our knowledge, only a limited subset of the meteorological variables used in this paper are available in the HRES analysis, which thus prevents a fully consistent comparison between the two meteorological products (HRES forecast and analysis).

Nonetheless, in order to avoid the confusion existing in the first version of the manuscript, we modified the corresponding sections by presenting this test HRES versus ERA5 as a simple sensitivity test on the meteorological input data :

**L99 : "2.1.3 IFS and ERA5 meteorological data**

[revised manuscript text omitted]

**We also modified the abstract :**

L13 : "When considering MOS methods relying on meteorological information and comparing the results obtained with IFS forecasts and ERA5 reanalysis, the relative deterioration brought by the use of IFS is minor, which paves the way for their use in operational MOS applications."  $\rightarrow$  "The MOS methods relying on meteorological data were found to provide relatively similar performance with two different meteorological inputs."

The graphical representation (Figures) needs to be improved. Choice of the colour range in maps and choice of colour in time series plot make it often impossible to discern the different data sets. Various aspects of Fig 3 remain unexplained.

Regarding the maps (Figs. 1 and 4 in the first version), the color range corresponds to the range of values obtained at the different stations (for both OBS and RAW, in order to allow a direct comparison). We suspect that the reviewer's comment is more directly related to the bottom panels (about exceedances of d1max above 90 ppbv) in which the color bar values range between 0 and 20 while most of the points shown on the corresponding maps are purple/blue. The reason is that some high values were hidden behind other overlapping points. We modified the plot to make it easier to read, and updated accordingly the legend by adding : "In order to limit the overlap, stations are here plotted by decreasing value and with decreasing size (lowest values with largest symbols but in background, highest values with smallest symbols but in foreground).". We also modified the color of the time series (Fig. 2 in

the first version) to make easier to read. Regarding Fig. 3, we greatly modified the discussion in the revised version.

**Please summarise the result of 3.2, 3.3 and 3.4 in tables. That will shorten the paper and make it possible to compare the different results more easily.**

We included in the Supplement several tables to show a subset of the evaluation results on continuous and categorical metrics for all these sensitivity tests. We also modified the corresponding figures in the Appendix so that to be consistent with the new version of Fig. 3 (fewer metrics). As described in another answer, we greatly shortened the discussion on the sensitivity tests (section 3.4).

**Some specific comments:**

Abstract: Please quantify the achieved improvements by MOS to replace or justify phrases such "can be substantially improved"

We added information on the improvement in terms of RMSE and PCC :

L11 : "Our results show that O3 forecasts can be substantially improved using such MOS corrections and that this improvement goes much beyond the correction of the systematic bias. Although it typically affects all lead times, some MOS methods appear more adversely impacted by the lead time. When considering MOS methods relying on meteorological information and comparing the results obtained with IFS forecasts and ERA5 reanalysis, the relative deterioration brought by the use of IFS is minor, which paves the way for their use in operational MOS applications."  $\rightarrow$  "Our results show that O3 forecasts can be substantially improved using such MOS corrections and that improvements go well beyond the correction of the systematic bias. Depending on the time scale and lead time, root mean square errors decreased from 20-40% to 10-30%, while Pearson Correlation coefficients increased from 0.7-0.8 to 0.8-0.9. Although the improvement typically affects all lead times, some MOS methods appear more adversely impacted by the lead time. The MOS methods relying on meteorological data were found to provide relatively similar performance with two different meteorological inputs."

*L*67-71 *A summary of the results of the paper is not required in the introduction* We removed these lines.

**L 98 mention forecast start time**

We modified the sentence L97 : "The CAMS regional forecasts are provided over 4 lead days, hereafter referred to as D+1, D+2, D+3 and D+4 (starting at 0 UTC)."

L 101 Flemming et al. 2015 is not a reference for the operational NWP forecast of ECMWF

We replaced this reference by the following link : https://www.ecmwf.int/en/forecasts/datasets/set-i

L 120 Please consider the general comment about NRT availability of observations As far as we know, EEA observations are usually available with a 2-hours lag. We included this information, as described in a previous answer. L Please clarify, if model output is required for the PERS and MA approach. Please use the model independent methods as reference (see general comments) and not as an other MOS variant.

The persistence (PERS) approach only depends on observations, and is now considered apart from the MOS methods (and used as a reference forecast for computing skill scores). The moving average (MA) method is a MOS method where the raw CAMS predictions are used together with the observations : the raw forecast on a given day is corrected by the mean difference between raw and observed concentrations one or several days before. The modifications of this part of the text are described in another answer (related to the reviewer's comment on the persistence method).

L 145 Can the choice of the length of the adjustment period (30 days) be substantiated ?

The choice of 30 days is arbitrary and motivated only by computational reasons, we extended the discussion related to this aspect :

L146 : "For computational reasons, both CDFs are updated every 30 days (although an update frequency of one single day would be optimal in a real operational context)."  $\rightarrow$  "For computational reasons, both CDFs are updated every 30 days (although an update frequency of one single day would be optimal in a real operational context). The choice of a 30-day update frequency only aims at reducing the computational cost of running all MOS methods at all stations during the 2-year period. In a real operational context, only one day would have to be run, which would allow increasing the update frequency up to 1 day, i.e., the CDFs would be updated every day ensuring that we are taking benefit from the entire observational dataset available at a given time."

L 155 KF and other method are based on unbiased linear estimates (BLUE) So, the biases are not addressed in KF theory in general. Please clarify.

To the best of our understanding, although biases are indeed not addressed in the KF theory, the application of KF as a MOS approach is specific in the sense that it takes the forecast bias itself as the state variable of interest. We reformulated part of this section :

L150 : "Over the last decades, the Kalman filter (KF) theory has found numerous applications in problems with different levels of complexity. In atmospheric sciences, it offers a popular frame for sophisticated data assimilation applications (e.g., Gaubert et al., 2014, Di Tomaso et al. 2017), but can also be used as a simple yet powerful MOS method for correcting forecasts (e.g., Delle Monache et al., 2006, Kang et al., 2008, De Ridder et al., 2012). A detailed description of the KF algorithm can be found in Appendix B (as well as in Delle Monache et al., 2006).

KF provides an efficient way of estimating the forecast bias based on past model and observation information. For a given day at a given hour, the forecast bias is computed as a weighted average of (1) the forecast bias estimated one day before and (2) the corresponding observed forecast bias. Each of these two terms is weighted according to the value of the so-called Kalman gain (Kt) that intrinsically depends on the so-called variance ratio (see Appendix B for more details). The value chosen for this internal parameter substantially affects the behaviour of the KF, and

thus the obtained MOS corrections." → "The Kalman Filter (KF) is an optimal recursive data processing algorithm with numerous science and engineering applications (see Pei et al., 2017 for an introduction). In atmospheric sciences, it offers a popular frame for sophisticated data assimilation applications (e.g., Gaubert et al., 2014, Di Tomaso et al. 2017), but can also be used as a simple vet powerful MOS method for correcting forecasts (e.g., Delle Monache et al., 2006, Kang et al., 2008, De Ridder et al., 2012). The KF-based MOS method aims at estimating recursively the unknown forecast bias (here taken as the state variable of interest) combining previous forecast bias estimates with forecast bias observations. The updated forecast bias estimate is computed as a weighted average of these two terms, both being considered as uncertain, i.e. affected by a noise with zero-mean and a given variance. A detailed description of the KF algorithm can be found in Appendix B but an important aspect to be mentioned here is that each of these two terms is weighted according to the value of the so-called Kalman gain that intrinsically depends on the ratio of both variances (hereafter referred to as the variance ratio). The value chosen for this internal parameter substantially affects the behavior of the KF, and thus the obtained MOS corrections."

Also, to make it easier to follow, we modified the corresponding Appendix and adopted notations more consistent with those used by Pei et al. (2017) in their gentle introduction to KF.

**L 180 Please clarify "best analogue days". How many days are required to get a spunup AN (10) method.**

The aforementioned distance metric is used to compute the distance between the current forecast day and each of the past days, this distance representing how similar or different are the current forecast day and one given past day (a small distance means that both are very similar). In this frame, best analog days refers to the most similar days (i.e. the days with smallest distance to the current forecast). We clarified the text :

L176 : "The current forecast is compared to past forecasts based on a set of features including the raw O3 mixing ratio forecast from the AQ model and the 10-meter wind speed, 2-meter temperature, surface pressure and boundary layer height forecast from the meteorological model. The similarity of each day of forecast is assessed using the distance metric proposed by Delle Monache et al. (2011) and previously used in Djalalova et al. (2015) (see the formula in Appendix C). As a first approach, we consider the 10 best analog days, hereafter referred to as AN(10); other values are tested in Sect. 3.4)."  $\rightarrow$  "The current forecast is compared to each individual past forecasts in order to identify which ones are the most similar. Based on a set of features including the raw  $O_3$  mixing ratio forecast from the AQ model and the 10meter wind speed, 2-meter temperature, surface pressure and boundary layer height forecast from the meteorological model, the distance metric proposed by Delle Monache et al. (2011) and previously used in Djalalova et al. (2015) (see the formula in Appendix C) is used to compute the distance (i.e., to quantify the similarity) of each individual past forecast with respect to the current forecast. Then, as a first approach, the 10 best analog days that correspond here to the 10 most similar past forecasts are identified (hereafter referred to as AN(10); other values are tested in Sect. 3.4)."

*L 209 Please motivate the choice of the 30 day training period.* See previous answer on the topic.

L 233 Missing here is a skill score that assess the forecast skill against the persistency forecast

**See previous answer on the topic.**

L 225-233 The amount of accuracy measures is overwhelming and the reader can not easily follow that. Please reduce the number of measures to a minimum and explain what specific characteristic of the forecast performance is quantified by that measure. Try to introduce a nomenclature (say upper case vs lower case, latin vs bold) for name of MOS methods and accuracy measures.

Although we understand it might appear overwhelming at first read, we do think it is useful to show such a comprehensive set of metrics to highlight different aspects of the forecast performance (while MOS results in the literature are often shown only with a very limited number of metric, typically only continuous). In the revised version, we simplified Fig. 3 (and the figures in the Appendix) by removing a few metrics, namely MB and RMSE (we only kept nMB and nRMSE), as well as the intercept and the base rate, which should make it slightly easier to follow. More importantly, we added a section where metrics are more clearly introduced (as previous mentioned in another answer), and we tried to interpret in more detail the discussion of these different metrics in the discussion of the results.

*L* 266 Please explain Fig 3 in more detail. What do the overlaying symbols mean (one per stations , forecast day ?).

We clarified the legend of Fig. 3 : "Figure 3. Statistical performance of RAW and MOS-corrected CAMS O3 forecasts for continuous metrics (top panels) and categorical metrics related to the exceedance of the target (intermediate panels) and information threshold (bottom panels). The different symbols depict results obtained at different time scales (h: hourly; d: daily mean; d1max/dd1max: daily 1-hour maximum; d8max/dd8max: daily 8-hour maximum). In each panel, results are shown for the different methods (each with a given color). The overlaying symbols of decreasing transparency show the results at the different lead days from D+1 (most transparent) to D+4 (most opaque). [...]"

We also extended the discussion of Fig. 3, as described in another answer.

*L* 277... Please provide the various accuracy measure in a table (also including the MOS results) for a better representation of the results. Done.

*L* 335-445 Please see my general comment on section 3.4 As described in another answer, we shortened this section.

L448 Using the ER5 data set as "truth" compared to the HRES NWP forecast does not make sense. The HRES analysis should be used for that. Because of different resolution and model cycle the two data sets are not consistent. Please avoid the term IFS for the forecast because both ER5 and HRES are produced with the IFS.; L446 ER5 and HRES will not differ in the number of assimilated observations, if anything HRES will be better.

As described in another answer, we modified this section to take into account the comments of the reviewer.

**L 484 Please provide quantitative information about the improvements.**

Given that numerous quantitative information was provided in the previous section, we do not think it is useful to provide again some quantitative information here. We rather prefer to keep this discussion for a general discussion around the use of MOS.

L 494 The skill of a forecast (in a scientific sense) is defined by the improvements w.r.t to a reference, which should be persistency in your case. How compares RAW and the MOS methods using RAW to PERS is a question that should be answered. See text book by D.S. Wilks, Statistical methods for atmospheric science.

As described in another answer, we take into account the comments of the reviewer regarding this aspect, and greatly modified the manuscript.

L517 The finding that MOS results (using RAW) were more sensible to forecast lead time than PERS is interesting. One would expect a strong impact of the lead time for PERS. Please elaborate a bit more. Do the forecast show a drift perhaps introduced by the initialisation with analysis (assimilating AQ surface information)

We are not sure to follow the reviewer on this comment. Among all the forecast methods, PERS is clearly the most strongly impacted by the lead time, while the impact of the lead time on RAW is found to relatively small (and we indicated in the text that it could be "potentially due to their relatively coarse spatial resolution"). The MOS methods are typically moderately impacted by the lead time, likely simply because they typically rely on both raw forecast and past recent observations. We added L517 : "The performance of the RAW forecasts was found to be only slightly sensitive to the lead day, but this sensitivity was substantially stronger with some MOS methods (although lower than for the persistence method)."

L 575 After all this long discussions, it would be good to still make a recommendation. Which MOS scheme performed overall best and would be recommended for operational implementation.

One key message of our study lies in the large variability of performance of the different MOS methods from one metric to another. Thus, it is not possible to conclude with a clear recommendation as it directly depends on what the user is most interested in, and in the specific case of O3 exceedances forecasts, the respective cost of false positive and false negative predictions. However, through its results and sensitivity tests, our study provides a rich and useful material to help users to make their decision (although any MOS implementation requires testing different MOS methods and/or configurations as results obtained here with CAMS ensemble forecasts over Spain might evidently differ with another model and/or in a different region).

L 575 Please provide reference for GHOST

Although it is still in preparation, the reference for GHOST is already provided in the list of references (Bowdalo, D.: Globally Harmonised Observational Surface Treatment: Database of global surface gas observations, in preparation).

**Reviewer #3**

The paper entitled "Model Output Statistics (MOS) applied to CAMS O3 forecasts: trade-offs between continuous and categorical skill scores" by Petetin et al. is well written and provides a very interesting perspective on different statistical tools and machine learning approaches that can be used to improve air quality forecasts. It falls within the scope of ACP and I truly enjoyed reading it. The analysis is sound and truly comprehensive. I have only a few minor comments, that the authors may want to consider to improve the manuscript further.

In line 85 the authors say that in this study, daily mean, daily 1-hour maximum and daily 8-hour maximum are computed only when at least 75% of the hourly data are available (i.e. 18 over 24 hours). Theoretically speaking a day during which the data from 9 am to 4 pm is missing could qualify this criterion, yet both the computed d1max and d8max of such a day would be far off. Instrumental interventions such as service visits, purging with zero air to get moisture out of the system and calibrations usually occur during working hours. Hence the authors may want to consider applying a filter directed at daytime rather than night-time observations next time. This may remove a few extra data points but would have been preferable considering the target of the paper. The current choice is, however, hardly going to impact the results pertaining to the model evaluation of the various techniques with respect to d8max and d1max. The days analysed are driven mostly by observational stations reporting d8max >60 ppb or D1max>90. Hence most days included in the analysis would have daytime data. Nigh time events with d8max >60 ppb and d1max >90 ppb are relatively rare, although they do occur occasionally at high altitude stations. So, in my opinion the best way out without redoing the analysis would be to run a quick check on the data for the following two parameters

How many days with large data gaps during the day (9 am to 4 pm) were included in the analysis?

How many of the observed d8max and d1max events are night time events? Both numbers would be small and can be reported and discussed as limitation.

Considering all Spanish stations and all days in 2018-2019 with at least 18 over 24 hourly values available, we checked how many had at least 6-hour data gaps occurring between 8 and 15 UTC. In total, the frequency of such large daytime data gaps is only 167/314,005 (0.05%), of which only 12 are exceedances of the target threshold (over a total of 13,221), and 0 are exceedances of the information threshold. Checking how many days and stations have at least 4-hour data gaps occurring between 8 and 15 UTC, the total frequency increases to 1854/314,005 (0.6%), of which only 77 are exceedances of the target threshold and 0 are exceedances of the information threshold. Therefore, the situation of large data gaps during daytime indeed occurs, but very rarely, and thus should not impact significantly our results. We added some elements of information regarding this

point : L83 : "In this study, daily mean, daily 1-hour maximum and daily 8-hour maximum (hereafter respectively referred to as d, d1max and d8max) are computed only when at least 75% of the hourly data are available (i.e. 18 over 24 hours). Note that despite such data availability criteria, large data gaps at some stations and during some days might occur mainly during daytime (for instance due to maintenance operations that typically occur during working hours). Considering all stations and days with at least 18 hours of data, the frequency of data gaps exceeding 4 hours between 8 and 15 UTC was found to be only 0.6% (1854/314,005). Such situation occurs with a similarly low frequency on days exceeding the target threshold (77/13,221 or 0.6%) and never occurs on days exceeding the information threshold."

Figure 2: I find it hard to see the colour difference between the purple and black line. In particular where they are not superimposed. The colour contrast in Figure F1 which is similar is much better

We modified the color of this plot, and updated the caption :

"Figure 2. Time series of the mean O3 mixing ratios over the Iberian Peninsula, as observed by monitoring stations (in black) and as simulated by the CAMS regional ensemble D+1 forecasts (in purple yellow). Time series are shown at the hourly (h), daily mean (d), daily 1-hour maximum (d1max) and daily 8-hour maximum (d8max) time scales. O3 mixing ratios are averaged over all surface stations of the domain."

Figure 3: Some people have bad memory for abbreviations or the habit of skipping to the figures. Just like the authors gave the full form for (h: hourly; d: daily mean; d1max/dd1max: daily 1-hour maximum; d8max/dd8max: daily 8-hour maximum) which is much appreciated can they please give the full form of the abbreviations S, H, F, FB, SR, CSI, PSS, AUC, PCC (which people may be more familiar with a R) in the figure caption. It will save a lot of readers from having to scroll back to the method section where these are defined.

We added more information in the caption (note that following the recommendation of the other reviewer, we removed some of the metrics to make the plot easier to follow) :

"Figure 3. Statistical performance of RAW and MOS-corrected CAMS O3 forecasts for continuous metrics (top panels) and categorical metrics related to the exceedance of the target (intermediate panels) and information threshold (bottom panels). The different symbols depict results obtained at different time scales (h: hourly; d: daily mean; d1max/dd1max: daily 1-hour maximum; d8max/dd8max: daily 8-hour maximum). In each panel, results are shown for the different methods (each with a given color). The overlaying symbols of decreasing transparency show the results at the different lead days from D+1 (most transparent) to D+4 (most opaque). Metrics : normalized Mean Bias (nMB in %), normalized Root Mean Square Error (nRMSE in %), Pearson correlation coefficient (PCC), slope (unitless), normalized Mean Standard Deviation bias (nMSDB in %), Hit rate (H), False alarm rate (F), Frequency Bias (FB), Success Ratio (SR), Critical Success Index (CSI), Peirce Skill Score (PSS), Area Under the ROC Curve (AUC). See Sect. 2.4 and 2.5 for details on time scales and metrics, respectively."

**Other modifications**

- We harmonized the manuscript to use exclusively American English (e.g. "behavior")
- We added at L245 : "Over the Iberian Peninsula, annual mean O3 mixing ratios"
- L237 : "Ozone pollution over Iberian Peninsula and raw CAMS forecasts" → "Ozone pollution over Iberian Peninsula"
- Abstract L7 : "A key aspect of our study is the evaluation, which is performed using a very comprehensive set of continuous and categorical metrics at various time scales (hourly to daily), along different lead times (1 to 4 days), and using different meteorological input data (forecast vs reanalyzed)." → "A key aspect of our study is the evaluation, which is performed using a comprehensive set of continuous and categorical metrics at various time scales, along different lead times, and using different meteorological input datasets."
- Abstract L16 : "However, they are not necessarily the best in predicting the highest O3 episodes, for which simpler MOS methods can give better results." → "However, they are not necessarily the best in predicting the peak O3 episodes, for which simpler MOS methods can achieve better results.
- L37 : "As these MOS methods often significantly reduce systematic errors, bringing mean biases close to zero, they are also commonly referred to as bias-correction or bias-adjustment methods, although they may not aimed at reducing directly this specific metric. MOS methods relying on local data (first and foremost the local observations) can also be seen as so-called downscaling methods as they allow capturing some of the local features that cannot be reproduced at typical CTM spatial resolution." → "As these MOS methods often significantly reduce systematic errors, bringing mean biases close to zero, they are also commonly referred to as bias-correction or bias-adjustment methods, although they may not be aimed at reducing directly this specific metric. MOS methods relying on local data (first and foremost the local observations) can also be seen as so-called at the local observations) can also be seen as the local data (first and foremost the local observations) can also be seen as bias-correction or bias-adjustment methods, although they may not be aimed at reducing directly this specific metric. MOS methods relying on local data (first and foremost the local observations) can also be seen as so-called downscaling methods since they allow capturing some of the local features that cannot be reproduced at typical CTM spatial resolution."
- L113 : "Applying MOS in a worse case scenario of operational-like conditions"
  → "Applying MOS under restrictive operational conditions"
- L114 : "A novel aspect of this study is that it provides a comparison of a set of MOS methods under a worse case scenario of operational-like conditions, which can be described through two assumptions:" → "A novel aspect of this study is that we provide a comparison of a set of MOS methods under potentially restrictive training conditions in operational context. To mimic such restrictions we assume that"
- L119 : "On a given day, all MOS methods can only rely on the historical data accumulated so far." → "On a given day, the MOS methods can therefore only rely on the historical data accumulated since the beginning of the period. Our approach consists in understanding the behaviour of the

different MOS methods in a worse case scenario where a new or upgraded operational AQ forecasting system is implemented together with a MOS module for which there is little or no hindcast data."

- L121 : "As it will be described in more detail in the next section, some MOS methods require very limited prior information to achieve their optimal performance, while other need a larger amount of training data." → "As described in detail in the next section, some MOS methods require very limited prior information to achieve their optimal performance, while others need a larger amount of training data."
- L189 : "In this study, we also explore the use of ML algorithms as an innovative MOS approach for correcting AQ forecasts." → "We also explore the use of ML algorithms as an innovative MOS approach for correcting AQ forecasts."
- L358 : "Therefore, for detecting exceedances, considering PSS and/or CSI as the most relevant metrics (Appendix E), the PERS method shows its best performance for a time window of 1 d."
- L557 : "In this study, we considered a relatively short 2-year dataset but using a longer training dataset would likely require to build specific methodologies to tackle this issue, either by identifying and discarding the potentially outdated data, or by giving them a lower weight in the procedure." → "In this study, we considered a relatively short 2-year dataset but using a longer training dataset would likely require building specific methodologies to tackle this issue, either by identifying and discarding the potentially outdated data, or by giving them a lower weight in the procedure."